# Quantitative interactome of a membrane Bcl-2 network identifies a hierarchy of complexes for apoptosis regulation

Stephanie Bleicken[1,2,3,5], Annika Hantusch[4], Kushal Kumar Das[3], Tancred Frickey[4] & Ana J. Garcia-Saez [1,2,3]

The Bcl-2 proteins form a complex interaction network that controls mitochondrial permeabilization and apoptosis. The relative importance of different Bcl-2 complexes and their spatio-temporal regulation is debated. Using fluorescence cross-correlation spectroscopy to quantify the interactions within a minimal Bcl-2 network, comprised by cBid, Bax, and Bcl-xL, we show that membrane insertion drastically alters the pattern of Bcl-2 complexes, and that the C-terminal helix of Bcl-xL determines its binding preferences. At physiological temperature, Bax can spontaneously activate in a self-amplifying process. Strikingly, Bax also recruits Bcl-xL to membranes, which is sufficient to retrotranslocate Bax back into solution to secure membrane integrity. Our study disentangles the hierarchy of Bcl-2 complex formation in relation to their environment: Bcl-xL association with cBid occurs in solution and in membranes, where the complex is stabilized, whereas Bcl-xL binding to Bax occurs only in membranes and with lower affinity than to cBid, leading instead to Bax retrotranslocation.

[1] Max Planck Institute for Intelligent Systems, Heisenbergstr. 3, 70569 Stuttgart, Germany. [2] German Cancer Research Center, Im Neuenheimer Feld 280, 69120 Heidelberg, Germany. [3] Interfaculty Institute of Biochemistry, Eberhard Karls University Tübingen, Hoppe-Seyler-Str. 4, 72076 Tübingen, Germany. [4] University of Konstanz, Applied Bioinformatics, Universitaetsstr. 10, 78457 Konstanz, Germany. [5] Present address: ZEMOS, Ruhr-University Bochum, Universitätsstr. 150, 44801 Bochum, Germany. Correspondence and requests for materials should be addressed to A.J.G-S. (email: ana.garcia@uni-tuebingen.de)

The proteins of the Bcl-2 family are key regulators of several cellular functions including mitochondrial dynamics and apoptosis[1–3]. They form a complex network with multiple, parallel interactions that regulates the permeabilization of the mitochondrial outer membrane (MOM). Once the membrane is perforated, cytochrome *c* is released, which is considered the point of no return in the cell commitment to death. Because the Bcl-2 network lies at the heart of apoptosis regulation and is linked to diseases like cancer, Bcl-2 proteins are attractive targets in drug development[3, 4].

The Bcl-2 family is classified into three sub-groups: Bax and Bak are proapoptotic and directly mediate MOM permeabilization by opening pores at the MOM. Prosurvival proteins like Bcl-2, Bcl-xL, and Mcl-1 promote cell survival by inhibiting their proapoptotic counterparts. The BH3-only proteins have evolved to sense stress stimuli and to promote apoptosis either directly by activating Bax and Bak or indirectly by inhibiting the prosurvival Bcl-2 proteins[1, 2].

In healthy cells, Bax is monomeric and shuttles continuously between the cytosol and the MOM[5, 6]. During apoptosis, it accumulates at the MOM and undergoes a conformational change that leads to membrane-insertion, oligomerization, and MOM permeabilization[7–12], which is accompanied by Bax assembly into a mixture of lines, rings, and arc-like structures[13, 14]. The active membrane-embedded conformation is suggested to form a clamp-like structure that remodels the membrane and stabilizes pores of tunable size[10, 15, 16]. Bax activity is regulated by other Bcl-2 members, including cBid and Bcl-xL. Bid is inactive in the cytosol until it is cleaved by caspase 8 into the active form cBid, which consists of two fragments: p7 and tBid[17, 18]. cBid translocates to the MOM and promotes Bax activation[8], as well as the insertion of Bcl-xL into the membrane[19–21]. Bcl-xL inhibits apoptosis via three incompletely understood modes (Fig. 1a). Mode 0 proposes that Bcl-xL shifts the equilibrium between membrane-bound and soluble Bax towards the soluble form[5, 6]. In Mode 1, Bcl-xL sequesters activator-type BH3 only proteins like cBid, and thereby prevents Bax activation[21, 22]. Mode 2 proposes inhibition by direct interaction of Bax and Bcl-xL. However, this is based on indirect evidence like co-immunoprecipitation, the use of chimeric proteins, or interaction-defective protein mutants[22–24]. In addition, Bcl-xL alters the way cBid and Bax remodel membranes[16].

Several models aim to explain how the Bcl-2 network controls MOM permeabilization. The indirect activation or de-repression model[25] implies that Bax is spontaneously active, unless it is bound to and inhibited by prosurvival Bcl-2 homologs. BH3 only proteins can compete with this interaction by binding to the prosurvival Bcl-2 family members, which releases Bax to induce MOM permeabilization. In contrast, the direct activation model[26, 27] proposes that Bax is inactive until it interacts with an activator-type BH3-only protein, like cBid, which triggers membrane insertion and the conformational change. The unified[22], the embedded together[28], and the hierarchical models[29] integrate the de-repression and the direct activation idea into one model.

To understand how the association between Bcl-2 members is orchestrated to regulate MOM permeabilization, a systems approach that provides detailed, quantitative understanding of the relative affinities between full-length Bcl-2 proteins, especially of their active, membrane-embedded forms, is necessary. Performing detailed interaction experiments in living cells is extremely difficult or impossible, due to at least four reasons: (i) the many interactions competing simultaneously, (ii) the difficulties to calculate protein concentrations in organelles of living cells, (iii) the presence of a mixture of different regulatory post-translational modifications, (iv) and the use of fusions to green fluorescent protein (GFP) or similar fluorescent proteins,

which due to their size could affect the function and interactions of the proteins of interest. To solve these limitations, we used here a bottom up approach based on a minimal interaction network composed of full-length cBid, Bax, and Bcl-xL that reproduces the functionality of the Bcl-2 family in vitro. Although the extrapolation to the physiological context needs to be done with extreme care, reconstituted systems allowed great advances in understanding the detailed molecular mechanisms of Bcl-2 function[26, 30, 31]. The main advantage of our approach is that it is chemically controlled so that the individual interactions between Bcl-2 members can be studied, whereas additional factors, as well as post-translational modifications are absent, or can be added separately when necessary.

Scanning fluorescence cross correlation spectroscopy (FCCS) allows to selectively detect interactions within membranes by removing signals from solution, which was not possible in earlier studies[22, 30, 32]. This is a critical advantage as cBid, Bax, and Bcl-xL constantly shuttle between soluble and membrane-bound conformations,[5, 6, 18] and both environments should be considered separately. By applying FCCS on soluble and membrane-embedded proteins, we show that the interactions within the Bcl-2 family are spatially regulated. Soluble Bax is monomeric, while upon membrane insertion, it associates into homo-oligomers and hetero-oligomers with cBid and Bcl-xL. In contrast, cBid/Bcl-xL hetero-dimers are detected in solution and membranes. Bcl-xL also forms homo-dimers in solution and its C-terminal transmembrane region modulates the preference for interaction partners. Moreover, we show that membrane-associated Bax recruits soluble Bax and Bcl-xL to the membrane. Bax self-recruitment is a feed-forward mechanism to enhance Bax activity, whereas Bcl-xL recruitment is inhibitory by reducing the size of Bax oligomers via direct interaction and by translocating Bax back into solution. Our findings demonstrate that no additional components are necessary for Bcl-xL-mediated retro-translocation of Bax, which, based on our data, might be driven by Bcl-xL homo-dimerization in solution. This work has implications for the understanding of the Bcl-2 signaling network in its natural context and supports a new model for the integration of Bcl-2 interactions during apoptosis regulation.

## Results

**The majority of Bcl-xL molecules are dimers in solution.** Here we used solution and scanning FCCS to quantify the concentrations, diffusion coefficients (*D*), and the interaction of cBid, Bax, and Bcl-xL (coupled to individual fluorophores) in solution and membranes. FCCS measures intensity fluctuations of fluorophores over time using the detection volume of a confocal microscope. On the basis of the intensity fluctuations over time auto-correlation (AC) and cross-correlation (CC) curves are calculated. The detection volumes of the two detection channels do not overlap perfectly, which affects the maximum CC detectable and as a result our CC values are slightly underestimated. The effect of channel cross talk and noise were calculated with free versions of the used dyes. In solution, the CC was below 2% ($\mu$: 1.4, $\sigma$: 0.7; see Supplementary Fig. 1) and values above 2.8% ($\mu \pm 2\sigma$, or 95% confidence) indicate protein interactions. More detailed information is given in Supplementary Methods and in ref. [33].

We quantified the homo-interactions and hetero-interactions between cBid, Bax, and Bcl-xL in solution (Fig. 1b, c). Bax showed no interactions with itself, cBid or Bcl-xL, indicating that it was present as a monomeric protein. In contrast, the small but significant positive CC of Bcl-xL labeled with red and green dyes indicated the formation of Bcl-xL homo-complexes, most likely

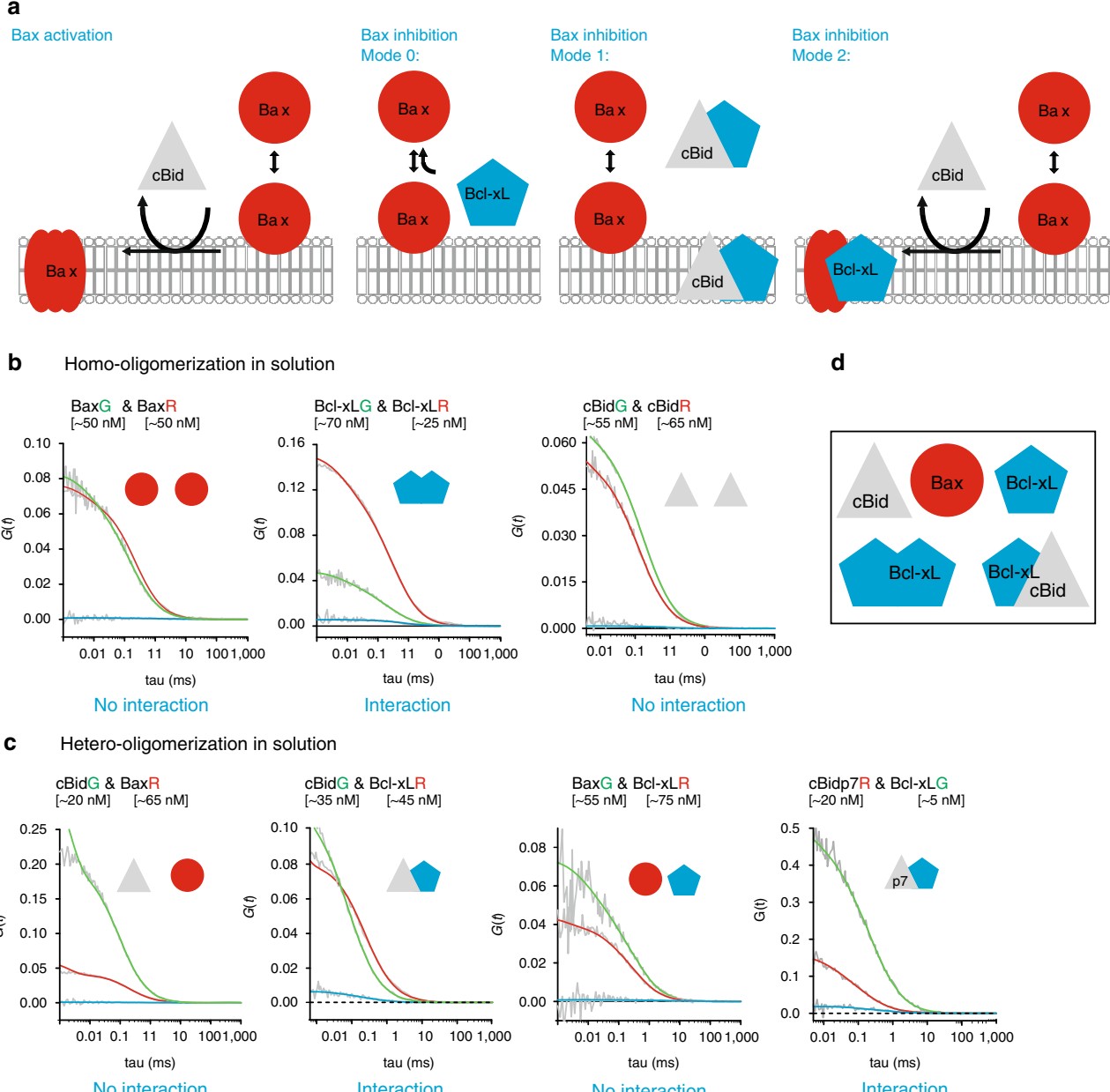

**Fig. 1** Analysis of interactions between cBid, Bax, and Bcl-xL in solution. **a** Representation of current models on Bax activation and inhibition by cBid and Bcl-xL. **b**, **c** Representative FCCS graphs of cBid, Bax, and Bcl-xL homo-hetero-oligomerization (B) and hetero-oligomerization in solution. For visual guidance: The amplitude of the AC curves is inversely proportional to the number of fluorophores, whereas the amplitude of the CC curves is proportional to the number of dual-color complexes. The decay of the AC and CC curves provides quantitative information on the diffusion properties of the particles and therefore of their size. All particle concentrations calculated from FCCS measurements refer to fluorescent particles that diffuse as a unit. Only Bcl-xL$_G$/Bcl-xL$_R$ and cBid$_G$/Bcl-xL$_R$ present positive CC amplitude (*blue curve*) indicative of interaction. The unfitted AC and CC curves are shown in *grey* ($n \geq 3$ independent experiments), and the fitted AC curves corresponding to the species labeled with green and red fluorophores and shown in *green* and *red*, respectively. We studied interactions at different protein concentrations up to 200 nM. **d** Scheme of the five different monomeric and dimeric species that can be detected in solution. *FCCS* fluorescence cross correlation spectroscopy

dimers, in line with ref. [34]. In addition, Bcl-xL interacted with cBid, as shown before[21, 32]. By using two cBid variants labeled at the N- (cBid-$_{p7R}$) or C-terminal fragment (cBid$_R$ or cBid$_{G,}$), we found that the cBid/Bcl-xL complex contained both cBid fragments. In addition, no homo-oligomerization of cBid molecules was detected. Our results show that in solution three different complexes are formed: p7/tBid, Bcl-xL/Bcl-xL, and p7/tBid/Bcl-xL (Fig. 1d). Those complexes are potentially competing in the cytosolic environment and based on our data only Mode 1 of Bcl-xL inhibition takes place in solution.

Contradictory results concerning the oligomeric state (monomeric vs. dimeric) of soluble Bcl-xL exist[34–39]. This is likely due to the use of C terminally truncated protein versions, which cannot dimerize[34]. Thus, the extent of Bcl-xL dimerization and its relevance remains obscure. The CC curves of Bcl-xL self-association showed low amplitudes, typical of weak interactions and supporting the existence of a major monomeric population (Fig. 1b, and Supplementary Fig. 2, $K_D$ ~600 nM). Intriguingly, the $D$ of Bcl-xL (Fig. 2a) was smaller than that of cBid and Bax, and varied strongly with protein concentration (Fig. 2a, b),

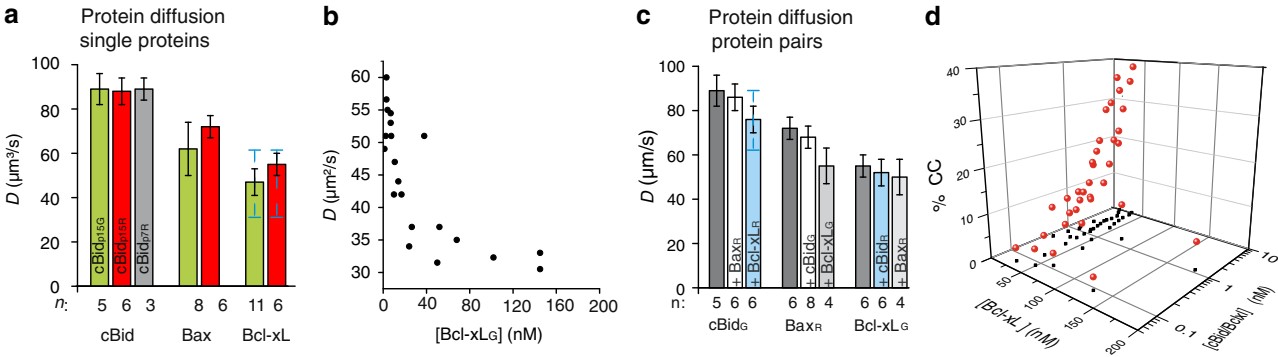

**Fig. 2** Quantitative analysis of the interactions between cBid and Bcl-xL in solution. **a** $D$ of cBid, Bax, and Bcl-xL labeled with Alexa488 or Atto488 (*green columns*), or Alexa633 or Atto655 (*red columns*). The *grey column* corresponds to cBid$_{p7R}$. Mean and error bars as s.d. are shown in black; the additional *blue error bars* for Bcl-xL indicate scattering of $D$ values depending on concentration, $n$ is indicated in the figure. **b** The $D$ of Bcl-xL$_G$ in dependence of the protein concentration ($n = 4$). **c** Effect of protein co-incubation on the $D$ of cBid, Bax, and Bcl-xL. Addition of Bcl-xL$_R$, induced a decrease in cBid$_G$ diffusion as expected from direct interaction ($n = 3$, mean and error bars as s.d. are shown in *black*); the additional *blue error bar* for cBid in presence of Bcl-xL indicates scattering of $D$ values depending on the Bcl-xL/cBid ratio. **d** The CC between cBid$_G$ and Bcl-xL$_R$ increased with protein concentration (data as *red dots* with *black shadows* in *xy*, $n = 6$)

suggesting a particle size in line with high dimer content. The structures of the inactive, soluble conformations of cBid, Bax, and Bcl-xL[17, 36, 40] show protein radii in the range of 1.7–2.5 nm, transferring into a $D$ of roughly 70–110 μm² s⁻¹. The $D$ of cBid, Bax, and Bcl-xLΔCT fall in this range (Fig. 2a and $D_{Bcl-xLΔCT}$: 78 μm² s⁻¹ see[21]), whereas the full length Bcl-xL had a smaller $D$ ($\sim 50$ μm² s⁻¹), as would be expected for Bcl-xL dimers. These results indicated that, full length Bcl-xL was mainly dimeric in solution, whereas the C-terminal truncated version was monomeric (in agreement with refs. [21, 34]).

To test whether we could detect full-length Bcl-xL monomers in solution, we diluted the protein (Fig. 2b). Interestingly, the $D$ of Bcl-xL grew at concentrations below 20 nM indicating an increase in the monomer population at lower concentrations in line with a high binding affinity between Bcl-xL monomers (low nano-molar $K_D$). In human cells, the concentration of Bcl-xL (10–1000 nM) and Bax (50–500 nM) was recently estimated[41], suggesting that at physiological concentrations Bcl-xL is mainly dimeric.

To further correlate particle mobility and interactions within the minimal Bcl-2 network, we tested how the presence of a second Bcl-2 protein affects the $D$ of cBid, Bax, and Bcl-xL (Fig. 2c). As expected from the CC data, Bax diffusion was not affected by the presence of cBid or Bcl-xL and vice versa, whereas the mean $D$ of cBid decreased in the presence of Bcl-xL, as expected for hetero-dimerization (p-value: 0.0089, unpaired two tailed t-test). In contrast, the $D$ of Bcl-xL was barely affected by the presence of cBid (Fig. 2c), in line with Bcl-xL being homo- or hetero-dimeric in solution.

**cBid can dissociate Bcl-xL homo-dimers.** Next, we probed the competition between Bcl-xL homo-complexes and the hetero-complexes by adding increasing amounts of cBid$_G$ to Bcl-xL$_R$ and measuring the CC after 1 h incubation. As expected for hetero-dimer formation, the amount of two colored complexes increased with the cBid concentration (up to 40% CC, Fig. 2d). Hetero-dimerization of this pair in solution was reported before[21, 22, 25, 32, 42–46], and in some cases quantified (with $K_D$'s: between 12 nM[42] and 350 nM[46]). However, this can only be an effective value, as it does not take into account the presence of Bcl-xL$_R$ homo-dimers.

To understand the interaction between cBid and Bcl-xL quantitatively, we needed to determine the exact reactions taking place. In Fig. 3a, three possible scenarios are suggested. In the

simplest scenario 1, cBid can only interact with Bcl-xL monomers and the formation of cBid/Bcl-xL and Bcl-xL/Bcl-xL complexes are competing. Scenario 2 differs from scenario 1 by an additional reaction, in which cBid is able to interact with Bcl-xL dimers, leading to the formation of a cBid/Bcl-xL hetero-dimer and one released Bcl-xL monomer. Scenario 3 considers instead that the interaction of cBid with a Bcl-xL dimer forms, a hetero-trimer, that can disassemble into two hetero-dimers after the addition of a second cBid molecule.

On the basis of these three scenarios, we designed an experiment that together with mathematical modeling allowed us to falsify one of the suggested scenarios. Bcl-xL$_R$ and Bcl-xL$_G$ were mixed and incubated for 120 min at room temperature (RT). Afterwards, the initial CC was measured (time −20 min) and the sample was divided into two equal parts. To one, we added unlabeled cBid in $\sim 10$-fold excess (time 0 min) to shift the equilibrium towards cBid/Bcl-xL complexes. To the second one, we added buffer as negative control. In both samples, the CC was followed over time (Fig. 3b). To our surprise, addition of cBid led to a clear increase in the CC between Bcl-xL$_R$ and Bcl-xL$_G$, whereas the CC remained unchanged in the negative control. Thus, cBid provoked not only hetero-dimer formation (Fig. 2d), but additionally boosted the amount to two-colored Bcl-xL homo-dimers (Fig. 3b). This can be explained by the existence of stable Bcl-xL homo-dimers with very slow exchange rates, so that in absence of cBid monomer exchange hardly takes place. cBid addition provokes hetero-complex formation and the release of Bcl-xL monomers, which in turn form new Bcl-xL homo-dimers increasing the number of two-colored Bcl-xL homo-dimers. Thereby, the total number of Bcl-xL homo-dimers did not increase and homo-dimer and hetero-dimer formation is in equilibrium.

**Modeling cBid and Bcl-xL interactions in solution.** To discriminate between the three reaction scenarios (Fig. 3a), we built mathematical models based on ordinary differential equations (ODEs), and analyzed the kinetics of association and dissociation of Bcl-xL molecules with themselves and with cBid. For scenario 1, a simple ODE system describing the two reversible interactions based on the law of mass action was fitted to the experimental data shown in Fig. 3b. The increase in two-colored Bcl-xL homo-dimers after cBid addition could not be reproduced by the model even when searching a large parameter space with a

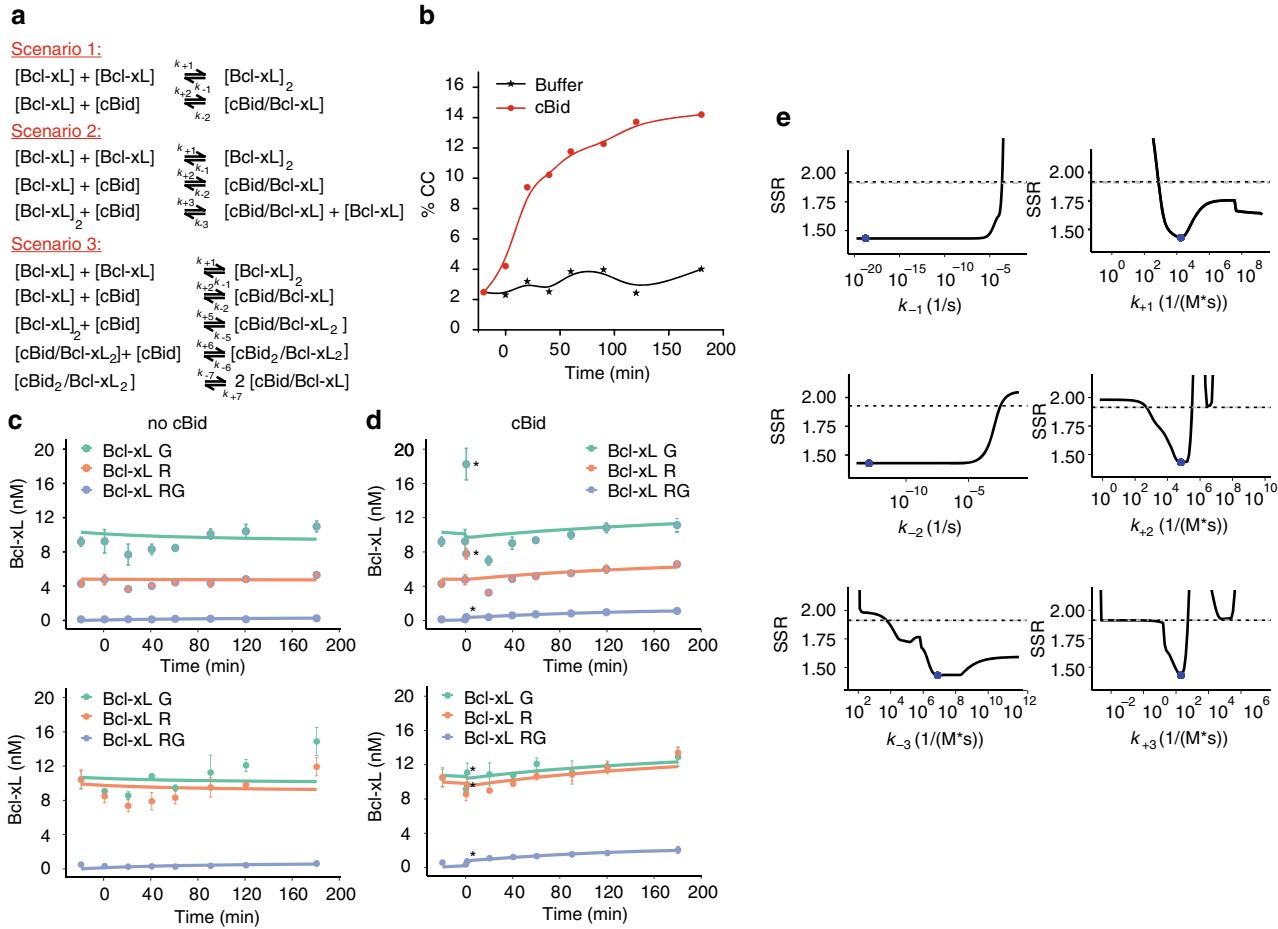

**Fig. 3** Bcl-xL forms stable dimers in solution that exchange in presence of cBid. **a** Possible reaction scenarios between cBid and Bcl-xL in solution tested by modeling. **b** FCCS kinetic experiment on Bcl-xL$_R$/Bcl-xL$_G$ dimerization in absence and presence of cBid (exemplary experiment out of $n = 3$). **c**, **d** Best fit of ODE model considering scenario 2 to the evolution of fluorescent Bcl-xL species based on the data shown in **b**. The *upper* and the *lower panels* show two independent experiments (mean and error bars as s.d. of three technical repetitions) with the *Y*-axis corresponding to the concentration of Bcl-xL particles. The first data point in presence of cBid is introduced as $t = 1$ min and highlighted in the figure by *asterisk*. This is applicable as each data point had a total FCS measurement time of 2 min. **e** Likelihood profiles of model parameters that define rate constants. The fitted parameter is shown as a *blue dot*, the reoptimized SSR is shown as *black line* and the *dashed grey line* indicates the confidence limit. *FCCS* fluorescence cross correlation spectroscopy

global optimization method (Supplementary Fig. 3). Thus, scenario 1 could be excluded.

In contrast, ODE modeling suggested that both scenarios 2 and 3 could quantitatively reproduce the experimental data (Fig. 3a–d and Supplementary Fig. 4). Thus, ODE modeling supports the idea that cBid can interact with Bcl-xL monomers and dimers.

To test whether scenario 2 or 3 are more likely, we used the difference in Akaike's information criterion (AIC), which considers the difference between the experimental data and the fit of the model as well as the number of parameters (model complexity). The differences in AIC values and the Akaike weights can then be used to select which scenario approximates the data best[47]. In our case, the values support scenario 2 over scenario 3 (Supplementary Table 2).

Finally, we performed a parameter identifiability analysis to examine and compare the velocity of all modeled association and dissociation reactions[48] (Fig. 3e and Supplementary Figs. 5 and 6). For scenario 2, two of the parameters were identified with 95% confidence within the tested parameter range, and the other could be constrained in at least one direction based on the 95% confidence limit and a reoptimized sum of squared residuals (SSR, with a minimum SSR value for each parameter). On the basis of the likelihood profiles of the model corresponding to scenario 2 (Fig. 3e), we could conclude that conversion of Bcl-xL homo-dimers to cBid/Bcl-xL hetero-dimers via cBid addition is a slow process ($k_{+3}$ 10–700 1/(M*s)). The slow association rate constant indicates that conformational changes are involved in the reaction leading to hetero-dimer formation[49]. A comparison of the likelihood profiles for $k_{-3}$ (exchange of cBid with Bcl-xL in a hetero-dimer to form a Bcl-xL homo-dimer) and $k_{+1}$ (association of two Bcl-xL) further suggested that cBid binding to Bcl-xL may speed up Bcl-xL dimerization. The calculated weak $K_D$ (~ 600 nM) for Bcl-xL homo-dimers (Supplementary Fig. 2) could then be explained by a very slow dissociation of Bcl-xL homo-dimers, causing the experimental system to reach an equilibrium much later than experimentally accessible. Thus the real $K_D$ for Bcl-xL homo-dimerization would be much smaller (in the order of 10 nM), as suggested before (Fig. 2b). In summary, our modeling data excluded scenario 1, whereas scenario 2 and 3 were both plausible with scenario 2 being more likely.

**In the membrane only Bax and Bcl-xL can self-associate.** The active conformations of the Bcl-2 proteins are membrane-embedded. Thus, it is crucial to understand the protein

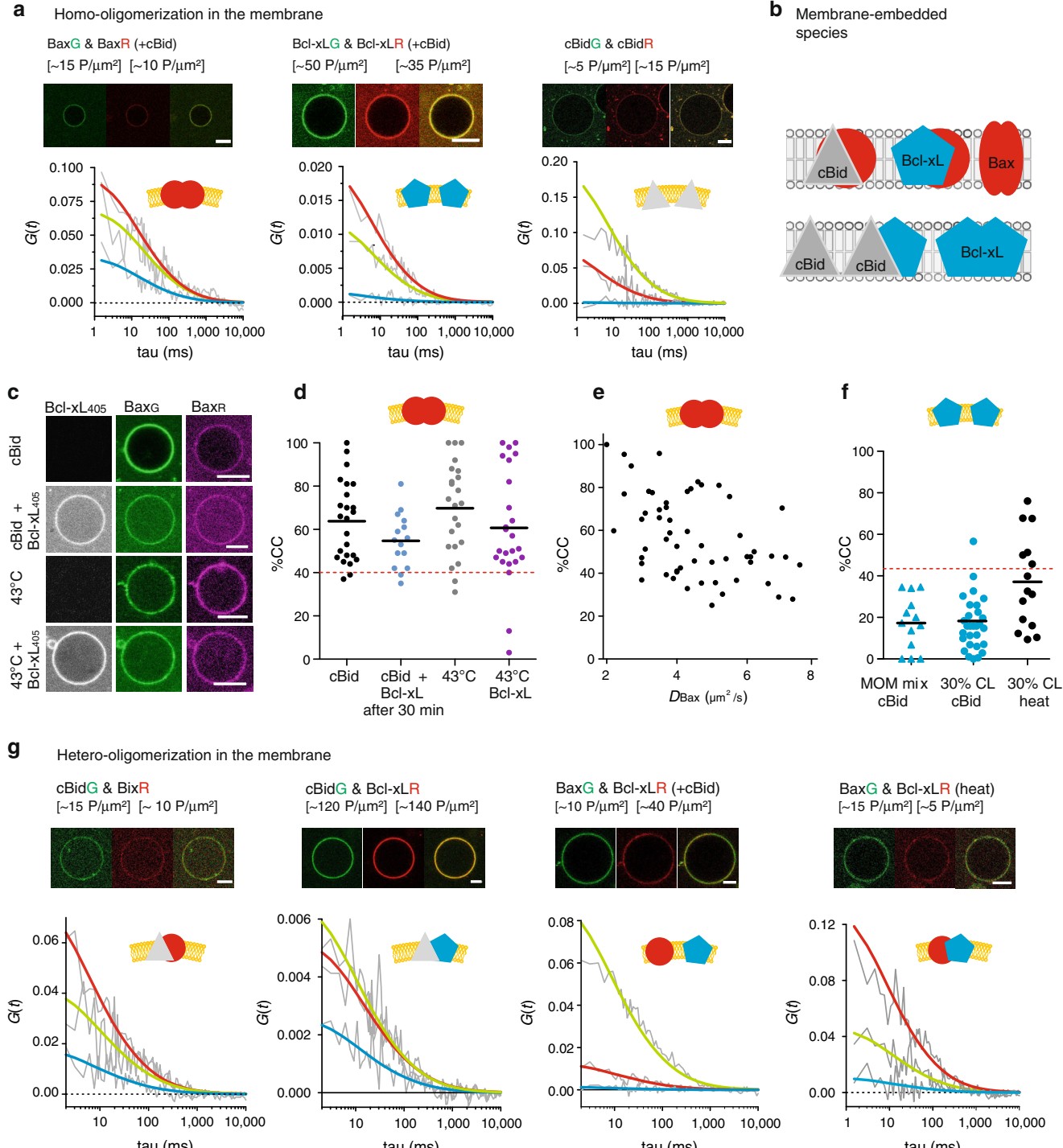

**Fig. 4** Analysis of the interactions between cBid, Bax and Bcl-xL within membranes. **a** Representative FCCS graphs and corresponding GUV images of cBid, Bax, and Bcl-xL homo-oligomerization in membranes (30% CL, Scale bar: 10 μm). Only the Bax$_G$/Bax$_R$ pair presents positive CC amplitude indicative of interaction. The AC and CC curves are shown in *grey*, whereas the fitted curves are shown in *green* (green protein), *red* (red protein), or *blue* (complex). **b** Scheme of the different species that can be detected in the membrane. **c** Representative images of Bax$_R$, Bax$_G$, and Bcl-xL$_{405}$ binding to GUVs (30% CL, Scale bar: 10 μm) in presence of unlabeled cBid or after incubation at 43 °C. **d** Comparison of the oligomerization expressed as %CC between Bax$_G$ and Bax$_R$ in individual GUVs after activation by cBid or 43 °C and in presence or absence of Bcl-xL$_{405}$ (30% CL; n = 4). All data are from one batch of labeled Bax to ensure that the degree of labeling is the same under all conditions. The maximal possible %CC for dimer formation considering the degree of labeling and a Mendel-based distribution of *red–red*, *red–green*, and *green–green* complexes is indicated by the *dotted red line*. **e** Relationship between CC and D of Bax$_G$ and Bax$_R$ activated by cBid in individual GUVs (30% CL; n = 5 independent experiments using three independent Bax batches; degree of labeling 80–100%). The CC values of three experiments were already published in ref. [19] (see also Supplemental Fig. 8A–C). **f** Comparison of the CC between Bcl-xL$_G$ and Bcl-xL$_R$ in GUVs composed of the both lipid mixtures. Membrane insertion was induced by cBid or 43 °C. The dotted red line indicates the maximal %CC possible considering dimer formation (n = 3). Only one protein batch was used due to the same reasons as in **d**. Data from independent Bcl-xL batches are shown in Supplemental Fig. 8A, B. **g** Representative FCCS graphs and corresponding GUV images of cBid, Bax, and Bcl-xL hetero-oligomerization in GUVs (30% CL; Scale bar: 10 μm; *color code* as in **a**. *FCCS* fluorescence cross correlation spectroscopy

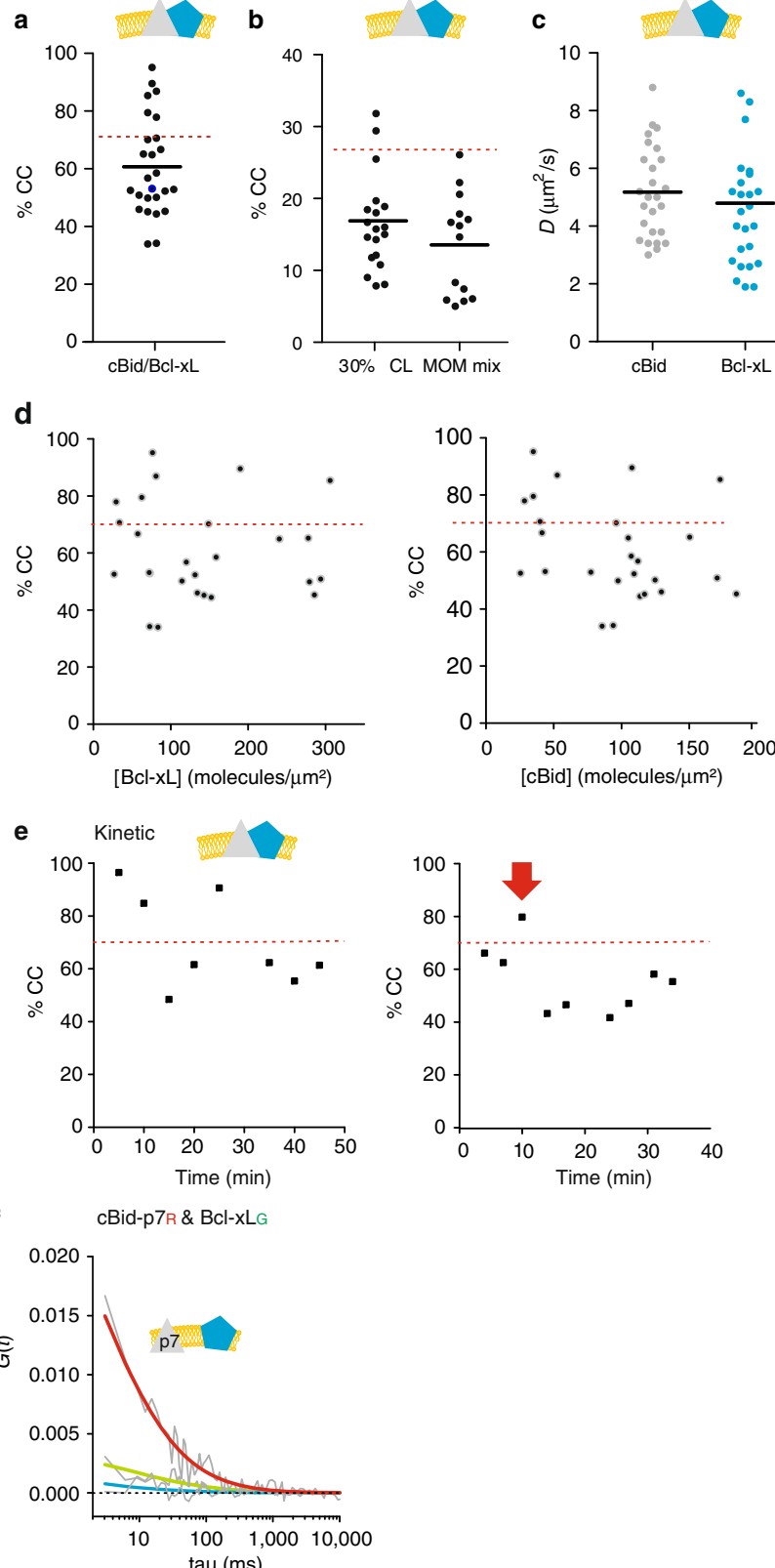

**Fig. 5** Quantitative analysis of cBid-Bcl-xL association in membranes. **a** %CC between cBid$_G$ and Bcl-xL$_R$ in membranes (30% CL; $n = 5$). **b** %CC between cBid$_G$ and Bcl-xL$_R$ in GUV membranes comparing both lipid mixtures ($n = 3$). **c** $D$ of cBid$_G$ and Bcl-xL$_R$ in individual 30% CL GUVs ($n = 5$). **d** %CC vs. the cBid$_G$ or Bcl-xL$_R$ protein concentration in GUVs (30% CL; $n = 5$). **e** Evolution of %CC between cBid$_G$ and Bcl-xL$_R$ in two individual GUVs as a function of time. The *red arrow* marks the time point at which the GUV was permeabilized (see also Supplemental Fig. 9). **f** Representative FCCS graph of cBid$_{p7R}$ and Bcl-xL$_G$ interaction. The absence of positive CC indicates that the labeled proteins do not interact in the membrane (*color code* as in 4 A). In **a**, **d**, and **e**, the dotted red line indicates the maximal CC considering dimer formation and degree of labeling. *FCCS* fluorescence cross correlation spectroscopy

interactions within the membrane. For this purpose, we measured scanning FCCS in giant unilamellar vesicles (GUVs) of two different lipid compositions (Methods and refs. [10, 15, 19, 30]): one is mimicking the MOM (MOM mix, ~ 5% cardiolipin (CL)), whereas the other has a high CL content (30% CL) to enhance protein membrane binding and therefore the contrast (Supplementary Fig. 7A). Both lipid mixtures have been studied before and Bax-induced membrane pores had similar properties in both[15], whereas permeabilization was more efficient with the higher CL concentration[15].

As a control for false-positive CC in the membrane, we measured the CC between Bcl-xL and a lipidic dye, two molecules that should not interact (Supplementary Fig. 7B). Based on this, we assumed that mean CC values above 20% (for 30% CL $\mu$+/−$2\sigma$ = 19% CC) indicated protein interactions in membranes. Some vesicles containing cBid and/or Bax exhibited bright spots, which we interpreted as membrane buds[16]. Those vesicles were excluded from analysis as artificial CC can be detected (Supplementary Fig. 7C). Finally, to induce Bax and Bcl-xL membrane insertion, we used two methods: we added cBid[19], or we applied a mild heat treatment[11, 50, 51] to avoid effects of unlabeled cBid on the interactions measured.

After membrane insertion, the interaction network between cBid, Bax, and Bcl-xL strongly changed. Once inserted into the membrane, Bax formed homo-oligomers (in agreement with refs. [10, 12, 19, 30, 52, 53]) irrespectively of the activation method (Fig. 4a, c, d and ref. [11]). In line with oligomer formation, the mean $D$ of membrane-embedded Bax was clearly smaller than the $D$ of cBid or Bcl-xL, whereas the CC between Bax molecules was much higher than the CC between cBid or Bcl-xL molecules (Fig. 4a and Supplementary Fig. 8A, B). Moreover, the mean CC between Bax molecules was higher and more disperse than expected for pure dimers (reaching 100% in some GUVs, Fig. 4c). This is in agreement with higher order oligomer formation and a broad distribution of oligomers sizes (Fig. 4e and Supplementary Fig. 8C).

We did not detect Bcl-xL homo-dimers in presence of cBid (Fig. 4a, f and ref. [19]), but when Bcl-xL membrane-insertion was induced by heat, we observed a significant amount of CC in line with homo-complex formation. This demonstrates for the first time that the membrane-inserted Bcl-xL can self-associate when no other interaction partners are present (Fig. 4f). However, cBid and Bax were preferred interaction partners over the self-interaction. Finally, we could not detect cBid homo-dimers in the membrane (Fig. 4a). This is at odds with studies reporting cBid homo-oligomerization in membranes[54]. One reason could be cBid ability to reorganize membranes[16], which can lead to artificial oligomer detection (Supplemental Fig. 7c).

Of note, we published CC data on Bax and Bcl-xL homo-interactions in membranes before[19]. Here, we included those data together with two new independent experimental repetitions, to compare the CC and $D$ values with other complexes, as well as measurements under different conditions, e.g., absence of cBid (data used already in ref. [19] are included in Fig. 4e, f and Supplemental Fig. 8a, c). In summary, membrane-embedded Bax and Bcl-xL could be detected in homo-complexes and hetero-complexes, whereas cBid was present as a part of hetero-complexes or as a monomer.

**In the membrane cBid–Bcl-xL complexes are stable and exclude p7**. The only complex detected in solution and membranes was the cBid/Bcl-xL hetero-dimer. Upon membrane insertion, hetero-complex formation dominated over Bcl-xL homo-dimerization (Figs. 4a, f, g and 5a), which indicates a stronger interaction between cBid and Bcl-xL in membranes

compared to solution. The mean CC between cBid/Bcl-xL in membranes was close to the maximum expected considering hetero-dimer formation (Fig. 5a), and it did not change significantly with lipid composition (Fig. 5b) or protein concentration (Fig. 5d). The mean $D$ for cBid and Bcl-xL was ~5 $\mu$m$^2$ s$^{-1}$ (Fig. 5c), in line with previous work[19, 21].

We also followed the kinetics of complex formation in individual GUVs (Fig. 5e and Supplementary Fig. 9) and found that the extent of association was high, stable over time, and independent of membrane permeabilization. This indicates again a very high binding affinity between cBid and Bcl-xL, and suggests that they insert into the membrane as a complex or, if there is recruitment, that it happens very fast. Thus, Mode 1 inhibition of MOMP will mainly happen in the membrane-bound state.

The soluble cBid/Bcl-xL complex contained tBid and the p7 fragment (Fig. 1c). To test if the same was true for the membrane-embedded complex, we used cBid$_{p7R}$. The CC between labeled p7 and Bcl-xL was as low as in the negative controls (Fig. 5f) indicating that the membrane-embedded complex does not contain p7.

**cBid–Bax interaction decreases upon Bax oligomerization**. cBid has been proposed to interact transiently with Bax to catalyze its activation and membrane insertion[12, 30, 31, 52, 55]. A transient interaction seems necessary as the binding site of Bid to the Bax BH-groove is overlapping with one interaction interface between Bax monomers in the homo-oligomer[12]. Detecting the cBid/Bax complex in membranes has remained challenging and could only be validated once[30]. Here, we could identify cBid/Bax complexes as well (Fig. 6a), but the level of CC was much lower than for cBid/Bcl-xL complexes (<40% CC compared to > 60% CC), indicating a low affinity or a short lifetime of the cBid/Bax complex.

We wondered if the dispersed CC data (Fig. 6a) could be the result of two populations of GUVs: one with high and one with low cBid and Bax interaction. On average, Bax clearly diffused slower than cBid (Fig. 6b), supporting this hypothesis. However, in GUVs with a high CC between cBid and Bax, cBid diffused similar to Bax (Fig. 6c), suggesting that Bax-oligomer assembly happened faster than cBid release. In addition, kinetic experiments revealed a decrease in the CC between cBid and Bax over time, in line with a transient interaction (Fig. 6d and Supplementary Fig. 10). Thus, the cBid–Bax complex in membranes is likely transient, and cBid is released after Bax homo-oligomerization.

**In membranes Bcl-xL hetero-dimerizes with Bax, but prefers cBid**. Although the inhibitory role of Bcl-xL via direct interaction with Bax (Mode 2) was first proposed more than two decades ago[56], the interaction between these two proteins has escaped detailed characterization. Here, despite the lack of association in solution (Fig. 1c), membrane-embedded Bax and Bcl-xL formed complexes (Fig. 6e). They were mainly detectable when membrane insertion was induced by mild heat treatment. In the presence of cBid, we could hardly detect Bax/Bcl-xL complexes (Fig. 6e–g and Supplementary Fig. 8D), likely due to the formation of competing cBid/Bcl-xL complexes (Fig. 5a). Performing these experiments was complicated as Bcl-xL excludes Bax from membranes[5, 6, 19]. To overcome this difficulty, we incubated cBid, Bax$_G$, and GUVs for 30 min prior to Bcl-xL$_R$ addition, which allowed Bax to insert into the membrane before Bcl-xL was added. This experiment revealed two important facts. First, a direct interaction between Bax and Bcl-xL is possible, but takes place only in the membrane-bound state.

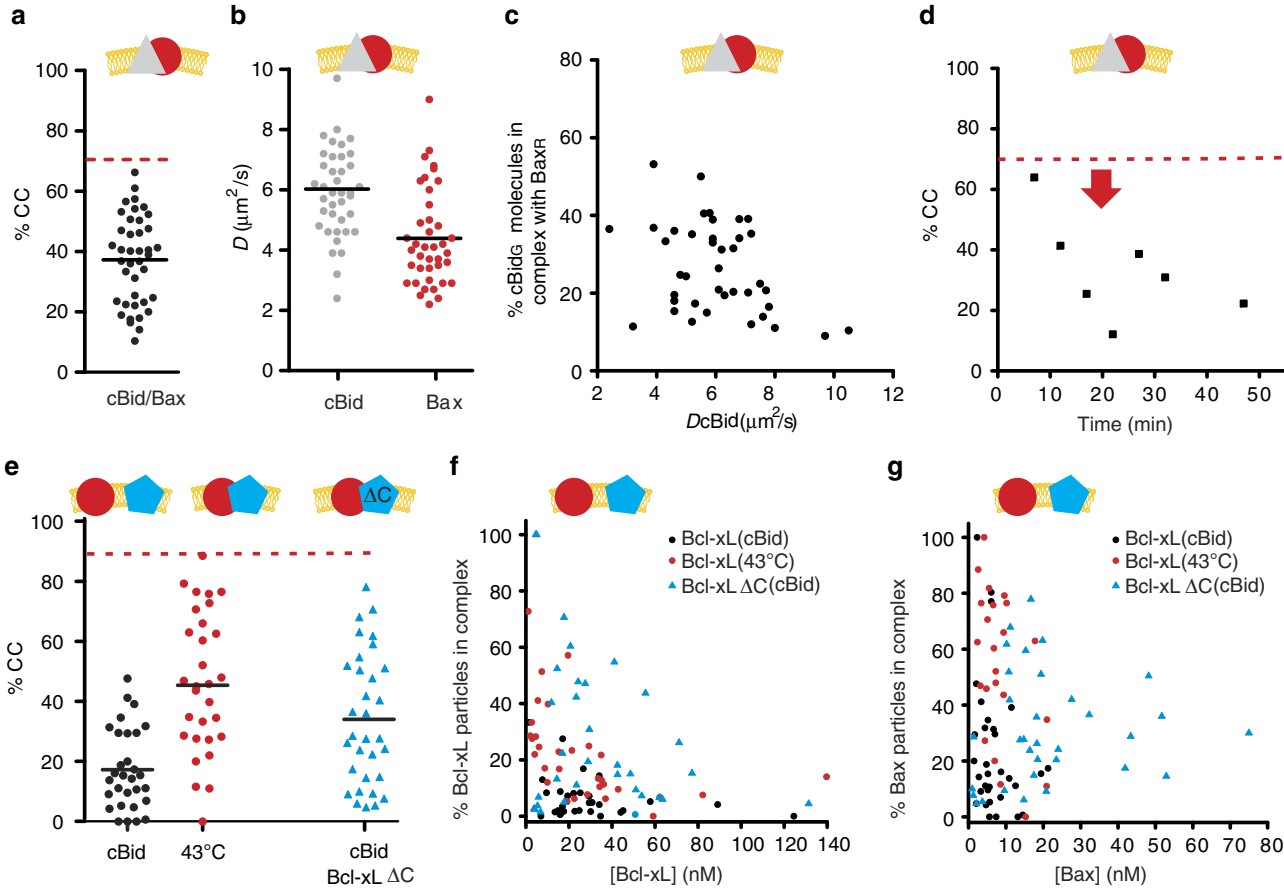

**Fig. 6** Quantitative analysis of the interaction of Bax with cBid and Bcl-xL in membranes. **a**, **b** %CC (**a**) and D (**b**) of membrane-embedded cBid$_G$ and Bax$_R$ in individual GUVs (30% CL; $n = 8$). **c** Percentage of cBid$_G$ molecules in complex with Bax$_R$ vs. the D of cBid$_G$ in individual GUVs (30% CL; $n = 8$). **d** Temporal evolution of %CC between cBid$_G$ and Bax$_R$ in a single GUV. The *red arrow* marks the time point at which the GUV was permeabilized (see also Supplemental Fig. 9). **e** Comparison of %CC between membrane-embedded Bax$_R$ (90% labeled) and Bcl-xL$_G$ (100% labeled) or Bcl-xLΔCT$_G$ (90% labeled) in individual GUVs (30% CL). Protein insertion into the membrane was activated by cBid or heat, as indicated ($n = 3$). **f** Percentage of membrane-embedded Bcl-xL or Bcl-xLΔCT molecules in complex with Bax in relation to the absolute concentration of Bcl-xL (after activation by cBid (*black dots*) or heat (*red dots*)) or Bcl-xLΔCT (*blue triangles*) ($n = 3$). **g** Percentage of membrane-embedded Bax molecules in complex with Bcl-xL (after activation by cBid (*black dots*) or heat (*red dots*)) or Bcl-xLΔCT (*blue triangles*) in relation to the absolute concentration of Bax ($n = 3$). The *dotted red line* in **a**, **d**, and **e** indicates the maximal possible CC considering the degree of labeling and dimer formation ($n = 3$). **e–g** Experiments done with one protein batch so that the results can be directly compared

Second, Bcl-xL interacts more strongly with cBid, when both partners are present, which indicates that Bcl-xL has a higher affinity for cBid than for Bax.

**The presence of Bcl-xL reduces the size of Bax oligomers.** These results raised the question of the functional impact of Bcl-xL on Bax oligomerization in membranes, which we addressed by comparing the CC between membrane-embedded Bax$_G$ and Bax$_R$ in presence and absence of Bcl-xL (Fig. 4c, d). To visualize Bcl-xL binding to GUVs, we used Bcl-xL$_{405}$ (labeled with Alexa 405), which was imaged in a third detection channel. The experiment was done after inducing Bax and Bcl-xL membrane insertion either with heat or cBid. When heat was used, Bax$_R$, Bax$_G$, and Bcl-xL$_{405}$ were added simultaneously to the GUVs, whereas when cBid was used, Bcl-xL$_{405}$ was added after Bax oligomer formation. In both cases, Bcl-xL$_{405}$ decreased the average CC between Bax molecules, indicating that Bcl-xL inhibited Bax oligomerization or reduced the oligomer size (Fig. 4d and ref.[11]). This suggests that Bcl-xL is able to bind to the membrane-embedded, active conformation of Bax.

FCCS is an equilibrium method that cannot establish the order of events for single molecules or the stoichiometry within complexes. Thus, we cannot distinguish whether Bcl-xL binds to and disassembles large Bax oligomers or whether it preferentially binds to monomers or dimers, preventing them from forming larger oligomers. However, we observed that membrane-embedded Bcl-xL diffused slightly faster in presence of cBid than in presence of Bax (Supplementary Fig. 8E), which suggests that Bax/Bcl-xL hetero-complexes are bigger as cBid/Bcl-xL dimers. This supports the idea that Bcl-xL is able to bind Bax oligomers, and suggests that one Bcl-xL molecule is able to inhibit more than one Bax molecule. However, we cannot discard the possibility that Bcl-xL preferentially binds Bax monomers.

**Bcl-xL C-terminus regulates the preference of interactions.** Until recently, most of the work done with recombinant Bcl-2 proteins used truncated proteins without the C-terminal membrane-anchoring helix. We tested the implications of this truncation on Bcl-2 interactions, because this helix has been related to homo-complex and hetero-complex formation,[10, 34, 57, 58] and

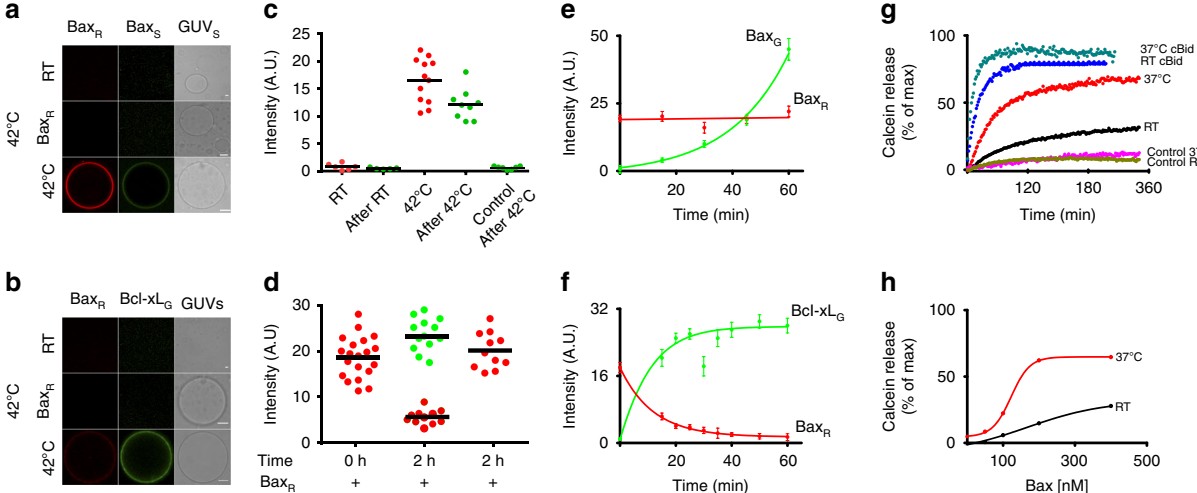

**Fig. 7** Bax auto-activation and retrotranslocation in model membranes. **a** $Bax_G$ binding to GUVs (30% CL) with or without membrane-embedded $Bax_R$ at room temperature (RT) and after incubation at 42 °C for 30 min. (Scale bar: 10 μm). **b** Binding of $Bcl-xL_G$ to 30% CL GUVs containing or not $Bax_R$ at RT and 42 °C for 30 min. **c** Quantification of $Bax_R$ (red dots) and $Bax_G$ (*green dots*) binding to individual GUVs before and after incubation at RT or 42 °C. The control corresponds to samples treated at 42 °C but in the absence of $Bax_R$. (*Black line* represents mean of the distribution). **d** Quantification of $Bax_R$ (*red dots*) and $Bcl-xL_G$ (*green dots*) binding to individual GUVs after heat activation (0 h after Bcl-xL addition) and after 2 h incubation at RT in presence and absence of Bcl-xL. (*Black line* represents mean of the distribution). **e** Temporal evolution of $Bax_R$ (previously bound by incubation at 42 °C) and $Bax_G$ intensity on GUVs at RT. (Error bars represent s.d. of $n = 3$ experiments). **f** Temporal evolution of $Bax_R$ (previously bound by incubation at 42 °C) and $Bcl-xL_G$ intensity on GUVs at RT. (Error bars represent s.d. of the average intensity of all GUVs in three experiments). **g** Kinetics of calcein release from LUVs, comparing 200 nM Bax activity at ~37 °C and RT in the presence and absence of 20 nM cBid. **h** Dose response curve of % calcein release at varying Bax concentration at ~37 °C as compared to RT. For all experiments $n = 3$

removal of the helix interfered with homo-dimerization[34] as well as with Bax retro-translocation[59, 60]. To do so, we compared the interaction of $Bax_R$ with full length $Bcl-xL_G$ and $Bcl-xL\Delta CT_G$ (Fig. 6e–g). Similar to the full-length version, $Bcl-xL\Delta CT_G$ did not interact with $Bax_R$ in solution (data not shown). However, in contrast to full-length Bcl-xL, the truncated protein failed to inhibit Bax membrane insertion, and once in the membrane, it interacted with $Bax_R$ even in the presence cBid (Fig. 6e–g). This demonstrates that the C-terminal helix of Bcl-xL tunes the hierarchy of interactions with other Bcl-2 family members.

**Membrane-bound Bax can recruit soluble Bax and Bcl-xL.** After analyzing the interactions between Bax, cBid, and Bcl-xL in solution and in membranes, we examined the translocation between both environments. We took advantage of direct visualization of protein binding to GUV membranes and investigated protein recruitment and retro-translocation. The presence of all three proteins leads to cBid and Bcl-xL translocation to the membrane, whereas Bax stays largely in solution[19]. Recently, it was suggested that membrane-bound Bax recruits soluble Bax via an auto-activation mechanism[57]. To test this hypothesis, we incubated 30% CL GUVs with $Bax_R$ at 42 °C, inducing $Bax_R$ association to the membrane (Fig. 7a, c). After cooling down, we added $Bax_G$ molecules, incubated for 1 h at RT and imaged the vesicles. $Bax_G$ bound to GUVs, confirming that membrane-bound Bax can recruit soluble Bax molecules (Fig. 7a, c, e). Binding did not happen when $Bax_R$ was absent or not treated with heat. To our surprise, membrane-associated $Bax_R$ recruited not only soluble $Bax_G$ but also soluble $Bcl-xL_G$ to the membrane (Figs. 7b, d). The accumulation of $Bcl-xL_G$ on the membrane was accompanied by a decrease in the mean fluorescence intensity of membrane-bound $Bax_R$ (Fig. 7b, d), indicating that $Bcl-xL_G$ promoted the release of Bax molecules

from the membrane into solution. The kinetics of this association/dissociation processes are shown in Fig. 7e, f. Interestingly, Bax recruitment to the membrane increased with the amount of Bax associated to the membrane, supporting a positive feedback mechanism. In contrast, the association of Bcl-xL to the membrane was faster initially, when most Bax molecules were membrane-bound, and decreased with time as the concentration of Bax in the membrane decreased (Fig. 7e, f). Thus, membrane-bound Bax recruits Bcl-xL to the membrane without positive feedback or additional recruitment of Bcl-xL by Bcl-xL. Most importantly, our data show that Bcl-xL promotes the dissociation of Bax from the membrane in the absence of any additional component. To our knowledge, this is the first measurement of Bax retro-translocation in recombinant systems.

These findings suggest that the energy barrier for Bax binding to the membrane, which is the limiting step in activation, is low. Indeed, in absence of all other Bcl-2 proteins, low expression of Bax in cells leads to its spontaneous activation, characterized by accumulation at the MOM and cell death[61]. To determine the role of temperature in Bax activation, we performed vesicle content release assays at RT or 37 °C (Fig. 7g, h). Spontaneous activation of Bax was negligible at RT, whereas incubation at 37 °C induced significant permeabilization of the membrane even in absence of cBid. As expected, at both temperatures, addition of cBid led to full and faster membrane permeabilization. Altogether, these findings demonstrate that Bax can spontaneously activate at physiological temperature, which is amplified by a positive feedback mechanism.

## Discussion

Here we report quantitative analysis of the interactions within a minimal Bcl-2 network that takes into account the spatial regulation of complexes in solution and membranes. One important finding is that the association of Bcl-2 proteins changes

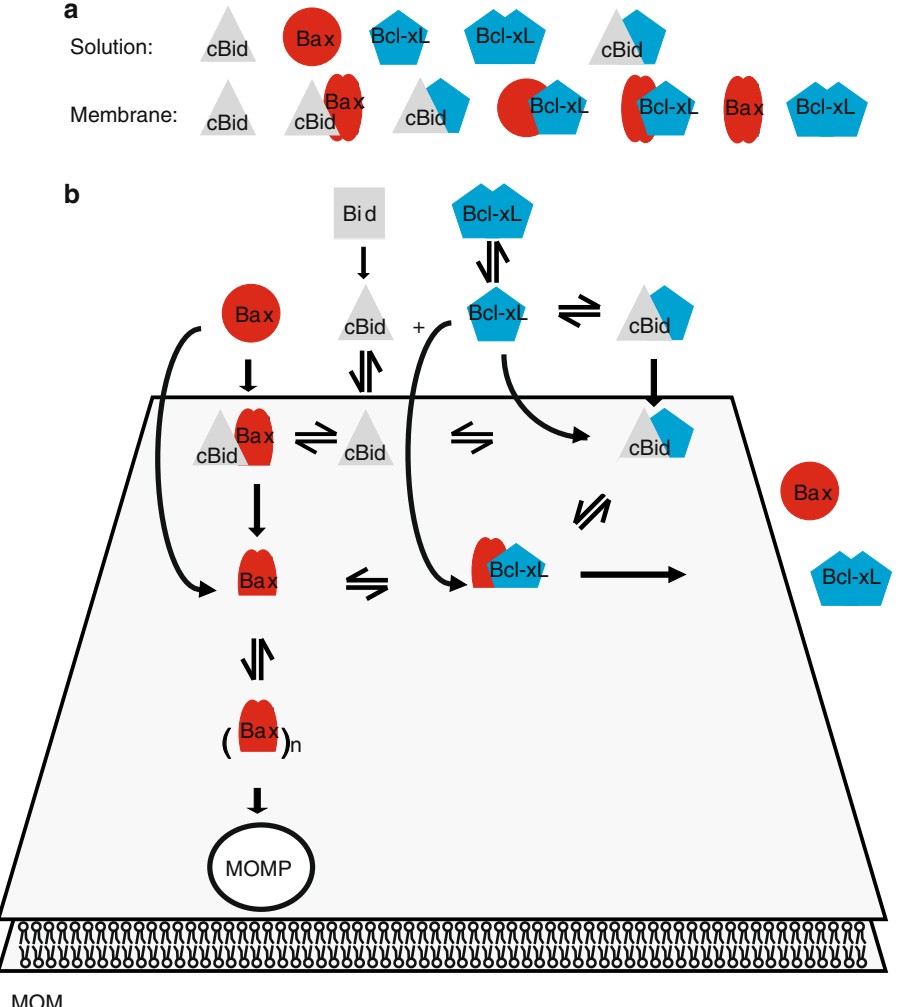

**Fig. 8** Integrated model for the multiple interactions between Bcl-2 proteins in solution and in membranes to regulate MOM permeabilization. **a** Species detected in solution and membranes. Bax is schematically shown in two conformations corresponding to monomeric as filled *red circles* and the membrane-embedded as *two red, filled ovals*. For the Bax/Bcl-xL complex, we showed that a complex containing oligomeric Bax exists, whereas we suggest the existence of the complex containing monomeric Bax. **b** Schematic representation of an integrated model of Bcl-2 protein interaction network in solution and at the MOM. *MOM* mitochondrial outer membrane

markedly upon membrane insertion, which is most likely due to the conformational changes associated with the process[10, 62]. In solution, Bax was monomeric, whereas cBid was present as a complex between its two fragments or associated with Bcl-xL. Bcl-xL itself hardly existed as a monomer, but formed homo-dimers and hetero-dimers with cBid (Fig. 8).

The existence of soluble Bcl-xL and Bax homo-dimers is strongly debated. Bcl-xL homo-dimers were detected in cells[34], but structural data are controversial[35, 37, 38]. Jeong et al.[34] showed that the C-terminal helix of Bcl-xL is critical for homo-dimerization and suggested that the helix of one monomer would bind to the hydrophobic groove of the second one, enabling dimerization. We found this hypothesis intriguing, as the C-terminal helix would be shielded in the dimer (Supplementary Fig. 10), which could explain why a fraction of Bcl-xL is cytosolic in cells despite the membrane anchor. Our data support this model, as we detected stable Bcl-xL dimers that dissociated by the addition of cBid. In this scenario, the C-terminal helix and the BH3 domain of cBid compete for the same binding site (the BH groove)[12], which provides a molecular basis for how cBid facilitates Bcl-xL membrane insertion by displacing the membrane anchor (Supplementary Fig. 10B).

In contrast to our results, the existence of soluble Bax homo-dimers and oligomers was recently reported[63, 64]. Garner et al.[63] detected autoinhibited Bax homo-dimers in the cytosol of the cell extracts of a number of cell lines, whereas in other cell lines, no Bax homo-dimers were found, suggesting that a very specific regulation might be necessary. Thus, the fact that we did not detect these dimers suggests that dimerization in solution requires additional factors (e.g., chaperones) or modifications absent in our system. Sung et al.[64] reported the oligomerization of soluble Bax upon incubation with BH3-only peptides at millimolar Bax concentrations. We never detected similar oligomers upon addition of cBid in the nanomolar (analyzed here) to micromolar[52] range. However, incubation of BaxΔC and BH3 peptides produced non-physiological swapped-dimers,[12] and we cannot discard that the soluble Bax oligomers in ref. [64] are multimers of swapped-dimers formed at very high protein concentrations.

Upon membrane insertion, the Bcl-2 interaction network strongly changed. Bax was always part of complexes with itself, with cBid, or with Bcl-xL, whereas Bcl-xL existed in complex with cBid or with Bax, or in the absence of both, as a mixture of monomers and homo-complexes (Fig. 8). This indicates that the

affinity of Bcl-xL to itself is lower than the affinity to cBid or Bax molecules. A transient interaction between cBid and Bax has been proposed as part of the "hit-and-run" model. Here, we provide additional evidence for the cBid/Bax complex and for its dissociation upon Bax homo-oligomization. This is in line with the fact that one Bax oligomerization interface overlaps with the binding site for cBid[12].

Despite the many models proposed for the regulation of Bax activation by the other Bcl-2 proteins, the lack of direct detection of Bax/Bcl-xL complexes in membranes and the lack of a quantitative characterization of the different interactions had left a key question open. When all proteins are present, does Bcl-xL inhibit apoptosis mainly by blocking the activators, like cBid, or the executioners, like Bax? Our results point to the first, as the affinity between Bcl-xL and cBid is larger than between Bcl-xL and Bax. This also explains why BH3 mimetics are efficient inducing apoptosis. Most likely they would not be as effective if Bcl-xL/Bax interactions were the stronger ones. Remarkably, our data showed that deletion of the C-terminal anchor of Bcl-xL altered its interaction preferences. This is likely due to the participation of the C-terminal helix in interaction surfaces[10, 57, 58]. Our data are at odds with the claim that Mode 2 inhibition is more efficient than Mode 1 inhibition in the unified model, but these studies where performed using C terminally truncated prosurvival Bcl-2 homologs[22]. Thus, caution should be exercised when interpreting experiments performed with truncated Bcl-2 members.

Our findings shed new light on Bax activation and inhibition. Recent work showed that in cells lacking all Bcl-2 proteins, Rb, and p53, reintroduction of Bax led to its spontaneous activation and cell death even at low Bax expression levels[61], in favor of the indirect activation model. However, the direct activation model is supported by the fact that Bax can be produced in vitro in an inactive form that can be activated by BH3-only activators to form membrane pores[15, 30], as well as by the structural data of BH3 peptides bound to Bax[12]. To reconcile both views, it is important to consider that activating Bax in absence of BH3 activators is also relatively easy: it can be achieved in vitro by exposure to mild heat[11, 51], acidic pH[65], detergents[26], or proteins like Drp1[66]. This all argues for a low energy barrier for Bax activation, in the order of thermal energy at physiological temperature. Here, we show that without activator molecules Bax remains inactive at RT, but presents a significant membrane permeabilizing activity at 37 °C. Once in the membrane, Bax promotes recruitment of soluble Bax. In the context of the cell, a small fraction of Bax molecules could spontaneously become active, which could be kept in place by complex formation with prosurvival Bcl-2 homologs. In agreement with this, we show that membrane-bound, active Bax-recruited Bcl-xL to the membrane and formed complexes with it, which led to a release of Bax from the membrane back into solution and to a reduction in Bax oligomer size. The protein retro-translocation into solution also supports a low energy barrier for dissociation of Bax/Bcl-xL complexes from the membrane. This barrier could be overcome by the high affinity between Bcl-xL monomers in solution, providing the necessary driving force for Bax retro-translocation. Moreover, the fact that membrane-bound Bax can recruit Bax and Bcl-xL supports a common molecular mechanism for both processes that was so far unanticipated, and it demonstrates that Bcl-xL is sufficient to retro-translocate Bax from the membrane back into solution.

Our observations link so-far unconnected observations: the low fraction of Bax/Bcl-xL complexes naturally found in mitochondria[29]; the finding that in cells lacking direct activators, Bax and Bak can still be activated and their activation depends on the absence of prosurvival Bcl-2 proteins[29, 61]; and the spontaneous

minority MOM permeabilization observed when prosurvival Bcl-2 proteins are inhibited with ABT-737[67]. It is important to consider that in cells, the Bcl-2 regulation network is much more complex and includes additional factors and post-translational modifications that add a layer of complexity to the observations reported here. Moreover, beyond their function in MOMP, Bcl-2 proteins have been associated with regulatory functions in mitochondrial dynamics and $Ca^{2+}$ homeostasis[1, 68–70]. The underlying mechanisms are not completely understood, but several Bcl-2 proteins are proposed to interact with MOM or ER membrane proteins without inducing MOM permeabilization[1, 68–70]. Therefore, the regulation of Bcl-2 protein function is more complex than considered so far, which could potentially also be examined with the system developed here.

On the basis of our findings, we propose a new, "integrated" model that explains how the multiple, parallel interactions between the Bcl-2 proteins are orchestrated to regulate apoptosis (Fig. 8). In absence of proapoptotic stimuli, Bax activation could proceed spontaneously. The membrane-bound Bax molecules could behave as seeding points for further Bax recruitment from solution in a positive feedback loop (here and refs. [11, 29, 61]). Direct interaction with prosurvival Bcl-2 homologs in the membrane would inhibit MOM permeabilization by relocation of loosely bound Bax molecules from the membrane to the cytosol, and it would additionally block the oligomerization of membrane-embedded Bax. In the presence of proapoptotic stimuli, the activation of BH3-only proteins would be first inhibited by direct association with the prosurvival Bcl-2s proteins, for which they have a higher affinity. In this scenario, a low level of Bax activation would still be counterbalanced by excess prosurvival Bcl-2s, but would prime the cells to die[27, 42]. On the basis of literature data, continued stress would further increase the levels of BH3-only proteins. Above a certain threshold in the relative concentration of proapoptotic vs. prosurvival Bcl-2 proteins, the prosurvival Bcl-2 homologs would be engaged mostly with BH3-only proteins and would not be able to sustain the steady state of Bax activation/inactivation, which would turn the balance and switch towards Bax activation and MOM permeabilization. In situations with intense cellular stress, any additional excess of BH3-only proteins would create a pool of free direct activators that would promote a rapid and efficient activation of Bax and Bak, full MOM permeabilization, and cell death.

In summary, the minimal network reported here explains the spatial regulation of Bcl-2 complexes in solution and membranes. We disentangle the hierarchy of competing reactions, as well as the modulatory role of the membrane and the C-terminal anchor, which has implications for the prevalence of the different inhibition modes depending on the environment and proteins present. At physiological temperatures, a fraction of Bax molecules spontaneously activates. These molecules can recruit to the membrane additional soluble Bax to promote MOM permeabilization, as well as soluble Bcl-xL, which inhibits Bax homo-oligomerization and releases Bax back into solution, thereby inhibiting MOM permeabilization. Altogether, these findings support an integrated model for Bcl-2 proteins that reconciles previously opposing experimental observations.

## Methods

**Protein production and labeling**. Full length mouse Bid (wild type, Bid C30S, or Bid C126S), full length human Bax (wild type and Bax S4C, C62S, C126S), and full length human Bcl-xL (wild type and Bcl-xL S4C, C151A) were expressed in *E. coli* BL21/RILP cells (Stratagene, now Agilent, Santa Clara, CA). Bacterial cultures were started at 37 °C at $OD_{600}$ ~ 0.03. Protein expression was induced with 1 mM IPTG at $OD_{600}$ ~ 0.5 followed by 4 h incubation at 20 °C. Cells were harvested by centrifugation at 6000 ×*g* for 20 min. The cell pellets were shock-frozen in liquid nitrogen and stored at −80 C. Before purification, the cells were thawed on ice,

resolved in buffer, and broken on ice by five passages though an Emulsiflex C5 (Avestin, Mannheim, Germany). Afterwards, ~1–200 U DNase I were added per liter bacterial culture (Merck, Darmstadt, Germany) and the mixture was incubated 30 min on ice. Then unbroken cells and membranes were removed by centrifugation at 25,000 rpm (60 min at 4 °C; using a JA25.50 rotor in a Beckmann Avanti centrifuge (Beckman Coulter, Brea, CA). Bid variants were purified using Nickel-NTA beads (Qiagen, Hilden, Germany) as the protein has an N-terminal His tag (plasmid pET23-His-Bid). Purification was done using 5 ml Nickel-NTA beads loaded into an empty gravity flow column. Buffers and purification steps were done following the manufacturer instructions, and protein elution was done by step a gradient adding 10, 25, and 250 nM Imidazol (in buffer). Protein purity was tested by SDS-PAGE showing about 95% purity. Afterwards to buffer was replaced by the caspase 8 cleavage buffer (50 mM NaCl, 5 mM DTT, 0.5 mM EDTA, 25 mM HEPES, 5% Sucrose; pH 7.4) using dialysis. Bid was cleaved to cBid by 4 h incubation with caspase 8 (at RT, Bid/Caspase 8 ratio ~1000:1; Caspase 8 was a gift J.-C. Martinou). Afterwards, a second purification (again based on Nickel-NTA) was done to remove Caspase 8, followed by an SDS PAGE performed to control protein purity (>95% see Supplemental Fig. 1E). Bax and Bcl-xL variants were expressed as intein-fusion proteins using the IMPACT-system from NEB (NEB, Ipswich, MA; plasmids pTYB1-BaxWT, pTYB1-Bax S4C, C62S, C126S, pTYB1-Bcl-xLWT, or pTYB1-Bcl-XL S4C, C151A). Buffers and purification were done following the manufacturer instructions. Samples were always kept on ice or at 4 °C and the cleavage reaction was done ~ 16 h at 4 °C. After elution protein purity was tested based on SDS-PAGE showing about ~ 90% purity. To remove residual impurities the sample was further purified using an anion exchange column (using a HiTrap Q column from GE healthcare on an AKTA purifier FPLC system from GE healthcare). First, the buffer was exchanged to 20 mM TRIS, pH 8 by dialysis to afterwards load the protein onto the column. The bound protein was washed with >20 column volumes (CV) of the buffer and then eluted with a gradient (8 CV) of high salt buffer (1 M NaCl, 20 mM TRIS, pH 8). The elution fractions were analyzed by SDS-PAGE (purity >95%, see Supplemental Fig. 1E) and finally, the buffer was exchanged using dialysis (to 150 mM NaCl, 20 mM TRIS, pH 7.5 or 150 mM NaCL, 20 mM HEPES, pH 7.5). The wild type proteins were aliquoted in 10 µl portions and shock-frozen in liquid nitrogen. Protein mutants were labeled before freezing (with Alexa 488-maleimide and Alexa 633-maleimide in the case of the cBid and ATTO 488-maleimide or ATTO 655-maleimide in the case of Bax and Bcl-xL). For labeling, the protein was incubated with threefold excess TCEP for 30 min on ice. Afterwards, a tenfold excess of the label was added and the sample incubated 2 h at 4 °C, before another fivefold excess of label was added and the sample incubated over night at 4 °C. Free label and protein were separated using desalting columns, and the degree of labeling was calculated using a combination of UV-VIS spectroscopy, Bradford assays, and ESI-LC-MS.

**Composition of the lipid mixtures.** The lipid mixture mimicking the MOM had a composition of 49% egg L-α-phosphatidyl choline (PC), 27% egg L-α phosphatidyl ethanolamine (PE), 10% bovine liver L-α-phosphatidyl inositol (PI), 10% 18:1 phosphatidyl serine (PS) and 4% CL (all percentages mol/mol). Moreover lipid mixtures composed of 30% CL and 70% PC or 20% CL and 70% PC (mol/mol) were used. All lipids were purchased from Avanti polar lipids (Alabaster, AL) and mixed in chloroform. Afterwards the chloroform was evaporated overnight under vacuum and then flushed with nitrogen or argon gas and stored at −28 C.

**GUV formation and sample preparation.** GUVs were produced by electro-formation and the experiments were done as described in ref. [19]. Briefly, 5 µg lipid mixture dissolved in chloroform were spread on each platinum electrode of the electro-formation chamber and allowed to dry, before immersion in 300 mM sucrose. Electro-formation proceeded for 2 h at 10 Hz, followed by 30 min at 2 Hz. Overall, 75 to 100 µl of the GUVs suspension was added to a solution of buffer mixed with the proteins of interest in Lab-Tek 8-well chamber slides (NUNC) to a final volume of 300 µl.

For experiments of Bax and Bcl-xL binding to GUVs, the sample mixtures were prepared in 8-well Lab-Tek chamber slides (NUNC) in buffer (150 mM NaCl, 20 mM Tris, pH 7.5) and 80 µl of GUV suspension, in total volume of 300 µl. The working concentration of $Bax_R$, $Bax_G$, and $Bcl-xL_G$ were 0.5–200 nM, respectively. To monitor Bax auto-activation, $Bax_R$ was incubated at 42 °C for 30 min, allowed to cool down to RT for 1 h, followed by subsequent addition of $Bax_G$ at RT and incubation for 1 h. The binding of the proteins to GUVs was imaged using a LSM710 confocal microscope. For Bax retro-translocation, $Bax_R$ was heat activated at 42 °C for 30 min, cooled down to RT for 1 h, followed by the addition of $Bcl-xL_G$ and incubation for 1 h at room temperature. To quantify the binding intensity of $Bax_R$, $Bax_G$, and $Bcl-xL_G$, the radial profile plugin of Image J was used with an integration angle of 60°. The background intensity was always taken into account for the intensity measurements. The curves were fitted using a nonlinear curve fitting function with sigmoidal dose response fit in Origin Lab.

**Calcein permeabilization assay.** LUVs composed of 80% PC and 20% CL were prepared by solving dried lipid mixtures in buffer (20 nM HEPES, pH 7.4 and 80 mM Calcein [fluorescein-bis-methyl-iminodiacetic acid at pH 7.5] with 4 mg mg⁻¹

lipid) using intensive vortexing paused by five cycles of freezing and thawing. The multilamellar vesicles were passed 31 times through an extruder (Avestin) using membranes with 400 nm pore size (Avestin). Calcein was entrapped in the vesicles at a self-quenching concentration, so that its release in external medium was accompanied by an increase of the intensity of fluorescence. LUVs were incubated with different concentrations of Bax, varying from 0 to 400 nM at ~ 37 °C in buffer (140 mM NaCl, 20 mM HEPES, 1 mM EDTA, pH 7.4) at room temperature with a lipid to protein concentration of >1:500 at the highest Bax concentration. The kinetics of calcein release were measured using a Tecan Infinite M200 microplate reader (Tecan, Männedorf, Switzerland).

The percentage of release R was calculated from the expression:

$$R = ((FS - F0) \div (Fmax - F0)) \times 100$$

where, F0 is the initial fluorescence of LUVs, Fmax is the maximum fluorescence after addition of 5% TritonX-100, and FS is the equilibrium fluorescence in the sample of interest.

**FCCS measurements.** All FCCS experiments were performed using a LSM710 confocal microscope equipped with a Confocor3, a C-Apochromat 40× N.A. 1.2 water immersion objective and laser to excite at 488 and 633 nm. Photons emitted from different fluorophores were separated by dichroic mirrors and detected by Avalanche photo diodes placed after suitable filters (for Atto/Alexa488, 505–540 nm band pass filter; for the far red dyes a >655 nm long pass filter). Each sample was measured at least 10,000× longer than their diffusion time to assure sufficient data points to generate the autocorrelation curves. To calculate the diffusion time ($t_D$), diffusion coefficients (D), protein concentration, and the cross-correlation, we assumed 3D Brownian diffusion and used the equations in Supplementary Table 1.

For solution FCS measurements, the proteins of interest were mixed with buffer (150 mM NaCl, 20 mM Tris, pH 7.5) in a total volume of 100–200 µl and incubated at least 30 min before measurements. Incubation and measurements were done in Lab-Tek 8-well chamber slides (NUNC) that were blocked with Casein (saturated solution in 150 mM NaCl, 20 mM TRIS, pH 7.5) before use. For all solution, FCCS measurements we did three technical repetitions and removed traces that contained large fluorescent particles disturbing the measurement. However, all n in the figure legends refer to experimental repetitions.

For scanning FCCS, we performed two-focus scanning FCCS measurements at 22 °C using a Confocor 3 module. Photon arrival times were recorded with a hardware correlator Flex 02-01D/C (http://correlator.com). We repeatedly scanned the detection volume with two perpendicular lines across a GUV equator (the distance between the two bleached lines d was measured on a film of dried fluorophores). Data analysis was performed with home-build software[21]. We binned the photon stream in 2 µs and arranged it as a matrix such that every row corresponded to one line scan. We corrected for membrane movements by calculating the maximum of a running average over several hundred line scans and shifting it to the same column. We fitted an average over all rows with a Gaussian and we added only the elements of each row between −2.5σ and 2.5σ to calculate the fluorescence intensity trace. We computed the auto-cross-correlation, spectral cross-correlation, and spatial cross-correlation curves from the intensity traces and excluded irregular curves resulting from instabilities and distortions. We fitted the auto-correlation and cross-correlation functions with a nonlinear least-squares global fitting algorithm as in ref. [21]. The equations used are shown in Supplementary Table 1.

In scanning FCCS, each value measured refers to one GUV. In total, we did n ≥ 3 independent experiments for each condition and interaction pair. In each experiment, we measured several GUVs that are shown as individual data points. Overall, 25–50% of measured GUVs were not included in the analysis for three reasons: (1) The GUV moved out of the focal volume during the measurement time (300 s). (2) Identification of large aggregates/buds on the surface of the GUV (see Supplemental Fig. 7) that strongly affected the measurement. (3) Large changes in protein concentration in the membrane during the measurement. The experiments were set up as a way that GUVs were added with a lipid to protein ratio of ~500:1 or higher.

**Mathematical modeling.** For each possible interaction scenario, an ODE-based model based on mass action kinetics was set up in COPASI (4.15, build 95)[71]. Differently labeled Bcl-xL proteins were included as separate species in the reactions.

Model parameters were fitted against four time courses of three different particle concentrations ($Bcl-xL_G$, $Bcl-xL_R$, and $Bcl-xL_G/Bcl-xL_R$) measured by FCS, with and without cBid addition. $Bcl-xL_G$ particles denote $Bcl-xL_G$ monomers and all possible multimers including at least one $Bcl-xL_G$ (same for $Bcl-xL_R$ particles). $Bcl-xL_G/Bcl-xL_R$ particles denote any particles including at least one $Bcl-xL_G$ and one $Bcl-xL_R$. Initial concentrations of monomeric and homo-dimeric $Bcl-xL_G$ and $Bcl-xL_R$ were included into parameter estimation (constrained between 0 and 10 nM).

For parameter fitting, first a global optimization method was applied by using the Evolutionary Programming algorithm in COPASI, where the population size was 10 times the number of parameters, and the maximum number of generations was 10 times the population size. To further improve the best parameter set found

by the global optimization method, the local method "Hooke and Jeeves" from within COPASI was used with default parameter settings. This procedure was described in ref. [72].

The following objective function was used for parameter fitting (weighted SSR, sum of squared residuals):

$$\text{SSR} = \sum_{i,j} \omega_j \cdot \left( x_{i,j} - y_{i,j}(P) \right)^2$$

with weight $\omega_j = \frac{1}{\langle x_j^2 \rangle}$; $j$ = time course of one particle type; $i$ = measurement at one time point; $x_{i,j}$ = measurement of one particle type at one time point; $y_{i,j}(P)$ = simulated value of one particle type at one time point given the parameter set $P$.

Bcl-xL$_R$ and Bcl-xL$_G$ were mixed and incubated 2 h at RT, before the first data point was taken at $t = -20$. Afterwards, the sample was split into two parts: to one buffer was added at $t = 0$, to the other 400 nM cBid was added at $t = 0$, thereby the sample was diluted 1:1. The dilution was accounted for by assuming half particle concentrations in measurements at $t = -20$ min. To improve fit performance, measurement of buffer control sample at 0 min was added to the time course including cBid addition, which was possible because the time courses of buffer control and cBid addition were derived from the same reaction sample and only separated at $t = 0$ min. The value of cBid addition was set from $t = 0$ to 1 min, which is justifiable by relatively long measurement times of FCS for every time point (2 min per time point).

The experiments were corrected for unlabeled Bcl-xL proteins by adding an unlabeled Bcl-xL species to the ODE systems. Initial concentrations of unlabeled monomers, and (partially) unlabeled Bcl-xL dimers were calculated the following way, assuming that the label has no influence on association and dissociation behavior:

DOL● -labeling efficiency of Bcl-xL$_G$ ($B●$); DOL○ -labeling efficiency of Bcl-xL$_R$ ($B○$)

unlabeled BclxL monomer:

$$[B] = (1 - \text{DOL}_{B○}) \cdot \frac{[B○]}{\text{DOL}_{B○}} + (1 - \text{DOL}_{B●}) \cdot \frac{[B●]}{\text{DOL}_{B●}}$$

unlabeled Bcl-xL homo-dimer:

$$[B-B] = (1 - \text{DOL}_{B○})^2 \cdot \frac{[B○ - B○]}{\text{DOL}_{B○}^2} + (1 - \text{DOL}_{B●})^2 \cdot \frac{[B● - B●]}{\text{DOL}_{B●}^2}$$

Bcl-xL homo-dimer including one labeled Bcl-xL (correspondingly for $B●-B$):

$$[B○ - B] = \text{DOL}_{B○}(1 - \text{DOL}_{B○}) \cdot 2 \cdot \frac{[B○ - B○]}{\text{DOL}_{B○}^2}$$

The standard deviation shown in the particle time courses of Bcl-xL$_G$–Bcl-xL$_R$ particles was calculated the following way: 10,000 numbers were sampled from the cross-correlation (CC) value of Bcl-xL$_G$ with Bcl-xL$_R$ with a gaussian distribution around the measured mean with the measured standard deviation. The same procedure was applied vice versa (CC of Bcl-xL$_R$ with Bcl-xL$_G$). From these 20,000 values the mean and standard deviation were calculated. All ODE equations used are listed in Supplementary Table 3.

Differences in AIC$_c$ (AIC corrected for small sample size) values and the Akaike weight $w_i$ were calculated as following and as described elsewhere[73]:

$k$ = number of parameters; $n$ = number of data points; $R$ = number of tested models

$$\text{AIC}_c = 2k + n \cdot \ln(\text{SSR}) + 2k \cdot \left( \frac{n}{n - k - 1} \right)$$

$$\Delta_i = \text{AIC}_{c,i} - \text{AIC}_{c,\min}$$

$$w_i = \frac{\exp(-\Delta_i/2)}{\sum_{j=1}^{R} \exp(-\Delta_j/2)}$$

**Parameter identifiability analysis.** Parameter identifiability analysis was performed and the 95% confidence regions of each parameter were determined as described in ref. [48]. Briefly, the objective function was reoptimized for each parameter on discrete logarithmic steps surrounding the optimized parameter value $\hat{p}$ with respect to all other parameters using the Hooke and Jeeves algorithm in COPASI. The confidence region was determined as described in ref. [48]:

$P_{\text{CR}} = \left\{ p : \text{SSR}(p) \leq \text{SSR}(\hat{p}) \left( 1 + \frac{k}{n-k} F^\alpha_{k,n-k} \right) \right\}$, where, $F^\alpha_{k,n-k}$ is the upper $\alpha$-critical value of the $F_{k,n-k}$ distribution.

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

## Acknowledgements

We thank C. Stegmueller for excellent technical assistance, J.C. Martinou for providing Caspase 8 and J. Suckale for critical reading of the manuscript. This work was supported by the Max Planck Society (S.B., K.K.D., and A.J.G.-S.), the German Cancer Research Center (S.B. and A.J.G.-S.), the German Ministry for Education and Research (BMBF, Grant No. 0312040; S.B. and A.J.G.-S.), the European Research Council (ERC-2012-StG 309966; S.B., K.K.D., and A.J.G.-S.) as well as the Forschergruppe 2036 (S.B., A.H., K.K.D.,T.F., and A.J.G.-S.), Konstanz Research School Chemical Biology (AH), and the Cluster of Excellence RESOLV (EXC 1069; S.B.) funded by the Deutsche Forschungsgemeinschaft.

## Author contributions

S.B. designed and performed experiments, analyzed data, and wrote the manuscript. K.K. D. performed experiments and analyzed data. A.H. and T.F. did the modeling and wrote the modeling part of the manuscript. A.J.G.-S. designed the study, planned experiments, and wrote the manuscript.

## Additional information

**Competing interests:** The authors declare no competing financial interests.

