## [Peer Review File · Nature Communications]

Reviewers' Comments:

Reviewer #1 (Remarks to the Author)

NCOMMS-16-15322

This refreshing study applies rigorous and defined microscopy, biophysical and mathematical approaches to interrogate the detailed interactions between the three best-studied (predominantly in mammals and cell culture) regulators of apoptosis. Many of the results are expected, but as expected when the details are delineated, there are also a few surprises.

Major comments:

1. Since submission of this manuscript, a new paper was published (Garner et al., Mol Cell 2016) reporting an inactive BAX dimer. Please discuss why these findings are seemingly inconsistent with the author's conclusion that BAX does not form soluble dimers, perhaps the type of dimer or other explanation.
2. As explained in the abstract and elsewhere, the authors conclude that BCL-xL acts directly on tunable oligomerized rings/arcs of BAX to reduce its size. This is mechanistically distinct from a dynamic BAX ring that liberates a molecule that is scavenged by BCL-xL, but the Results section was not sufficiently clear on this distinction. In most cases the text is beautifully articulated, but on this point, please better articulate the rigorous evidence that distinguishes these possibilities, or revise the language. Clarify whether or not BCL-xL is attacking and disrupting a formed (non-dynamic) oligomer versus catching it in its dynamic process.

Minor points/clarifications:

3. Please include in all the figure legends the protein concentrations or comparable metric. For example, why does it appear that the starting concentrations of the two proteins are different (difference between green and red lines) in some of the FCCS plots in Figure 1B, 1C, 4A, & 4F. Also indicate the number of independent experiments performed.
4. Why do some of the data points only in Figure 4C lie above the theoretical maximum?
5. In Figure 4A, is BCL-xL dimerization on GUVs altered in the absence of cBID?
6. Figure 5B – missing line for theoretical maximum?
7. Figure 7D – explain whether any of the data were collected in real time as implied, or not. Were the same GUVs monitored over time, or different populations of GUVs were used to generate the curve?
8. Figure 8 model – the 2-D trapezoid needs shading or head groups with an aspect ratio gradient to convey the presumably intended 3D images in the trapezoid is presumably intended to represent the surface of a mitochondrion. In addition, the lengths of the forward and reverse arrows could be adjusted to reflect high vs. low affinity interactions (i.e. cBID and BCL-xL).
9. Page 6 – confusion regarding labeling of the p7 versus tBID fragments. It appears that the experiment was performed with the p7 fragment only, rather than labeling this fragment within the monomeric p7-tBid complex. Please clarify.
10. End of first paragraph on page 10 – is the K_d of BCL-xL homodimer 10 nM with or without Bid?
11. Bottom of page 20 – there is a vaguely worded sentence: "Continued stress would further increase the levels of BH3-only proteins and decrease those of prosurvival proteins." Are the authors summarizing the field, expressing mathematical predictions or other? How are these levels regulated, transcriptionally, translationally, post translationally? Please explain or delete.

Reviewer #2 (Remarks to the Author)

The authors performed a systematic analysis of the interactions in the three-component system Bax, Bcl-xL and cBid, both free in solution and in artificial membranes. The aim was to understand the role of these proteins in the "hole-burning" of the mitochondrial membrane during apoptosis.

The paper seems to have been written in a hurry. The findings (which are mainly the associations of the different proteins under various conditions, as measured by FCS and FCCS) are listed in a lengthy results section, without attempting to summarize them into some scheme that helps the uninitiated reader understanding what is the main statement here.

The reader should not be left to the literature for understanding what scanning FCS actually is. The method, in particular two-focus scanning FCCS - and the procedure to correct for membrane movement during the measurement - must be briefly explained. It is certainly not textbook knowledge.

I would take exception to the claim that the system studied here is a "minimal interaction network" using a "synthetic biology approach". A network would consist of meshes, three partners would at maximum form a triangle, not really a network. There is also no "synthetic biology" in here, just classical biochemical / biophysical analysis of isolated protein-protein interactions. The authors should be careful with such buzzwords, which often are used only to make a study sound more "hip". Also the word "recruiting" should be replaced by "binding" unless there is any active process going on that is different from binding (by random diffusion).

In general, the language is typical of current "scientific paper jargon". As an example "multitude of" (p.4 last par.) could be substituted by "many" without loss of information, making lighter reading. "Bottom up synthetic biology approach" (same par.) conveys no information whatsoever, could be deleted. "Intriguingly, strikingly, dramatically, interestingly" and other overexaggerations should be deleted wherever they occur.

In several places there are incomplete phrases, sometimes rendering the text incomprehensible; e.g. p.7 last par.: "Next we probed the competitions between the Bcl-xL can form homo-dimers and hetero-complexes with cBid". ???

In spite of these language shortcomings, the Results section is well described, although the amount of detail presented without attempting to consolidate the findings into a framework makes it very difficult to read.

At any rate, the paper should be revised by a native speaker and good scientific writer.

Technical issues

1. The spectral ranges for the detection channels are nowhere indicated. This is critical for checking the consistency of the crosstalk/background figures given.
2. p.5, par.2: "the baseline for CC was below 2% (data not shown)" - this data **MUST BE SHOWN!** Supplementary information has a lot of space. Why is the threshold for (qualitative) interaction set to 3%, is this based on some statistical criterion or just arbitrary?
3. No positive control for 100% cross correlation is given. This could be easily checked with an A488 / A633 double labeled probe.
4. Since the positive control is missing, all statements regarding binding are necessarily qualitative. I do not understand how a detailed qualitative analysis of a complex thermodynamic model is possible in this case.
5. This is strikingly evident in Figure S1 (homodimerization). How can this graph be produced when the 100% binding limit is not known?
6. Association data between fluorescently labeled proteins, in particular the long-wavelength fluorophores, are notoriously prone to artifacts due to self-association of the dyes. This point is never discussed nor controlled.

7. How was the purity of the protein labeling checked (labeling degree and NO non-specifically associated fluorophore)?

8. p.8 last par.: the whole explanation of the increase of CC amplitude upon addition of cBid is completely obscure to me. Why would the formation of a cBid/Bcl-xL heterodimer lead to an increase of the CC amplitude between Bcl- xLR and Bcl-xLG?

Conclusion

In general, the paper reports on some interesting new findings, which lack, however, necessary controls and explanations. However, while the problem studied here is certainly interesting for people working in the field of mechanism of apoptosis, it seems to be too specialized to be of interest to a large multidisciplinary audience. I would not recommend publication in NCOMMS.

Reviewer #3 (Remarks to the Author)

Bleicken et al. present fluorescence cross correlation spectroscopy (FCCS)-based studies in solution and in a lipid membrane with mathematical modeling to systematically quantify the interactions within a minimal Bcl-2 network using representative proteins of the Bcl-2 family such as cBid, Bax and Bcl-xL. The study would require an independent review of the mathematical modeling and FCCS methodology, but also data on the purity, size and activity of the fluorescence labeled proteins, which are not provided.

Nevertheless, the study largely confirms interactions and conclusions previously presented in cell biology, structural biology and biochemical studies by other groups (several of these studies are cited in the paper).

For example:

- the associations of the proteins in solution and in membranes are known
- affinities are also established to be higher in the membrane than in solution
- spontaneous Bax activation by temperature, autoactivation mechanism and oligomerization in the membrane are also established
- recruitment of Bax and Bcl-xL in the membrane by cBID is previously shown
- inhibition of MOMP by Bcl-xL using a higher affinity to cBID than Bax is established
- dissociation of activated BAX in the membrane by Bcl-xL is established
- the key role of the c-terminal helix of Bcl-xL in regulating interactions is known

Also similar studies performed by the group of David Andrews which has found similar results in reconstituted membranes using FRET and biochemical readouts (Billel LP Plos Biol 2008, Lovell JF Cell 2008, Tan C JBC 2004, Shamas-Din A JBC 2013) Therefore, to mention that the "relative importance of the BCL-2 family interactions are poorly understood" is an overstatement. I understand that FCCS provides a fresh insight into the individual interactions of proteins compared to more established approaches but I also struggle to understand the impact of this study in more physiological context, where proteins can undergo modifications, receive stress signals and their concentrations change, which in several cases these phenomena cannot be fully reproduced by the recombinant proteins. Therefore, this study is more appropriate for a focused biophysical journal.

Reviewer #4 (Remarks to the Author)

In this well performed and clear study, Garcia-Saez and colleagues further develop our current knowledge of the BCL-2 family protein regulation of apoptosis, taking into account interactions within membranes. Through very careful experimental design and execution in an reconstituted systems (LUVs), the authors suggest that Bax can auto-activate (at physiological temperature) providing a positive feed-back mechanism that requires regulation by Bcl-xL. Bcl-xL regulation comprises three distinct steps identified by the authors: it translocates Bax back into solution, binds BH3 only proteins (cBid) limiting their interaction with Bax and binds Bax oligomers limiting their size. Each of these aspects is very important for our understanding of the process of MOMP,

and a 'unified' model has been developed by the authors.

Minor criticisms:

1. In the Discussion, the authors should discuss more carefully the limitations of their current system in terms of their reconstitution assays used.

2. The authors should also discuss how this model relates to Bax (and BCL-xL) activities OTHER than apoptosis regulation such as Ca²⁺ fluxes at the ER and fusion/fission regulation (which could be their primary functions evolutionary).

ANSWERS TO THE REVIEWERS

Reviewers' comments:

Reviewer #1 (Remarks to the Author):

NCOMMS-16-15322

This refreshing study applies rigorous and defined microscopy, biophysical and mathematical approaches to interrogate the detailed interactions between the three best-studied (predominantly in mammals and cell culture) regulators of apoptosis. Many of the results are expected, but as expected when the details are delineated, there are also a few surprises.

We thank the reviewer for these encouraging words and the positive evaluation of our work.

Major comments:

1. Since submission of this manuscript, a new paper was published (Garner et al., Mol Cell 2016) reporting an inactive BAX dimer. Please discuss why these findings are seemingly inconsistent with the author's conclusion that BAX does not form soluble dimers, perhaps the type of dimer or other explanation.

Following the reviewer's suggestion, we added now the reference and discussed the finding in relation to our results.

2. As explained in the abstract and elsewhere, the authors conclude that BCL-xL acts directly on tunable oligomerized rings/arcs of BAX to reduce its size. This is mechanistically distinct from a dynamic BAX ring that liberates a molecule that is scavenged by BCL-xL, but the Results section was not sufficiently clear on this distinction. In most cases the text is beautifully articulated, but on this point, please better articulate the rigorous evidence that distinguishes these possibilities, or revise the language. Clarify whether or not BCL-xL is attacking and disrupting a formed (non-dynamic) oligomer versus catching it in its dynamic process.

The reviewer raised a very interesting point. The FCS method we use here is an equilibrium technique that does not allow us to follow the chain of events of a single molecule. We cannot follow the exact event when Bcl-xL attacks a Bax oligomer or a released monomer/dimer. What we can address is the final outcome which is a reduction in the average size of Bax oligomers, as shown here and in our previous work, Subburay et al. (Nat Commun. (2015) 6:8042). Following the reviewer's suggestion, we have now rewritten the text to improve clarity.

Along these lines, here we gained not only information about the cross-correlation/interaction between molecules, but also about their diffusive behavior. This includes data on the Bax-Bcl-xL and cBid- Bcl-xL hetero-complexes in the membrane. Interestingly, Bcl-xL diffused slower in the presence of Bax as when cBid was present, indicating that the majority Bax-Bcl-xL complexes are bigger than dimers (see Figure 5A and Supp. Fig 8E). We believe that this provides information to the question raised by the reviewer. Accordingly, Supp. Fig 8E is now extended to show this effect better. Nevertheless, we cannot discard the existence of a sub-population of Bax-Bcl-xL hetero-dimers, as we measure the average diffusion coefficient per GUV. We have adapted the text to make this point clearer.

Minor points/clarifications:

3. Please include in all the figure legends the protein concentrations or comparable metric. For example, why does it appear that the starting concentrations of the two proteins are different (difference between green and red lines) in some of the FCCS plots in Figure 1B, 1C, 4A, & 4F. Also indicate the number of independent experiments performed.

The FCS curves from Figure 1 and 4 show exemplary curves of individual experiments to give the reader a visual indication whether the corresponding complex is formed or not. Following the reviewer's suggestion, we now added the protein concentration of the sample shown in each graph. We performed for all samples measurements at different concentrations between 2-200 nM with at least 3 technical repetitions in each of at least 4 independent experiments performed.

When not indicated otherwise, we tried to work with one-to-one ratios of the proteins and mixed them from the stocks according to their concentration calculated with standard biochemical methods (absorbance, Bradford, etc.). However with FCS, which also measures the concentration of fluorophores in the sample, we usually (here and also with the other proteins we have measured in our lab) find small differences, which are indeed shown in the amplitude of the FCS curves, as the reviewer points out. We believe this is due to two reasons: 1) differences in the results of the different methods to measure protein concentrations intrinsic to the experimental approach, 2) alterations in concentration due to protein handling. For example, proteins tend stick to glass or plastic, and the Bcl-2 proteins have an ever higher tendency because they are relative hydrophobic. To minimize the Bcl-2 protein loss to the chamber surfaces, we blocked these surfaces with casein directly before use. Unfortunately, we still lose some labeled Bcl-2 proteins to the chamber surfaces. As we work at nanomolar concentrations, the loss is enough to make it difficult to achieve exact one-to-one ratios.

4. Why do some of the data points only in Figure 4C lie above the theoretical maximum?

The lines indicate always the theoretical maximum of molecules that are both red and green considering the degree of labeling and assuming that only dimerization takes place. The exact stoichiometry of potentially formed oligomers is unknown, but we included this line to give a reference for the expected values in the case of dimers.

After the question of the reviewer, we realized that we made a mistake in placing the lines Figure 4D and F. In those two cases homo-complexes are formed. Thus, not only red-green dimers, but also red-red and green-green homo-dimers can be formed, which shifts the "line"(theoretical maximal CC) to lower values as shown in the new figures. We had considered this problem in our calculations already, but overlooked it when placing the line for visual guidance. We apologize for this mistake.

The measured cross-correlation in Figure 4D is now much higher as the "line" calculated for dimers, which make sense as Bax forms oligomers bigger than dimers. Therefore, the probability to have complexes with red and green molecules is much higher as considered for dimers. We adapted the text to make this point clearer.

In the case of Figure 4F, which refers to Bcl-xL, the correction for the position of the line does not affect our conclusions, as in presence of cBid, Bcl-xL still shows no clear sign of dimerization considering the level of cross-talk and noise in membranes. But now the difference between heat or cBid activated Bcl-xL becomes even clearer.

5. In Figure 4A, is BCL-xL dimerization on GUVs altered in the absence of cBID?

As shown in Figure 4F and mentioned above, in membranes and in presence of cBid, no clear Bcl-xL-homo-dimerization can be detected, as the data are close to the cross-talk/noise control. The data do not rule out the possibility that some Bcl-xL homo-dimers are formed in presence of cBid, but if they exist they are a minor population. This is most likely because most Bcl-xL molecules in the membrane are interacting with cBid in heterodimers. The situation indeed changes when heat is used to induce Bcl-xL's membrane insertion, as under those conditions we can clearly detect Bcl-xL homo-dimers in the membranes. Therefore as the reviewer comments, the dimerization of Bcl-xL in GUVs is affected by the presence of cBid. We have now adapted the text to clarify this point.

6. Figure 5B – missing line for theoretical maximum?

We thank the reviewer for the comment and added this information to the figure.

7. Figure 7D – explain whether any of the data were collected in real time as implied, or not. Were the same GUVs monitored over time, or different populations of GUVs were used to generate the curve?

The data were collected in real time and the same samples of GUVs were monitored over time. To generate the curve, several pictures were taken at different, random regions of each sample and therefore different populations of GUVs were used in the calculations. We have now clarified this in the text.

8. Figure 8 model – the 2-D trapezoid needs shading or head groups with an aspect ratio gradient to convey the presumably intended 3D images in the trapezoid is presumably intended to represent the surface of a mitochondrion. In addition, the lengths of the forward and reverse arrows could be adjusted to reflect high vs. low affinity interactions (i.e. cBID and BCL-xL).

We aimed to keep the model as simple as possible. Following the reviewer's suggestion we show now the lipid head-groups and try to improve the 3D aspect of the membrane. We have also added a panel to clarify that which complexes we detect in the membrane and in solution.

However, we prefer to not adjust the arrows length or thickness. We had already largely discussed this issue among the authors. The model aims to show a system in which all reactions happen "simultaneously", like in living cells. However, technically it is impossible to study all reactions at once. Our experimental setup allows us to study the preference for a given reaction at a time, in solution or in membranes, in presence or absence of additional components, and to follow recruitment or retro-translocation, but we cannot simultaneously follow all events. Although the reviewer's suggestion is very attractive, we prefer to avoid the risk of over-interpreting our data.

9. Page 6 – confusion regarding labeling of the p7 versus tBID fragments. It appears that the experiment was performed with the p7 fragment only, rather than labeling this fragment within the monomeric p7-tBid complex. Please clarify.

We never produced the p7 fragment alone. We labeled it within the cBid molecule. We adjusted the text to clarify this point.

10. End of first paragraph on page 10 – is the Kd of BCL-xL homodimer 10 nM with or without Bid?

The Kd was only assumed without cBid and the text is modified to clarify this.

11. Bottom of page 20 – there is a vaguely worded sentence: "Continued stress would further increase the levels of BH3-only proteins and decrease those of prosurvival proteins." Are the authors summarizing the field, expressing mathematical predictions or other? How are these levels regulated, transcriptionally, translationally, post translationally? Please explain or delete.

We thank the reviewer for this comment. We rephrased and reorganized the discussion to clarify this point.

Reviewer #2 (Remarks to the Author):

The authors performed a systematic analysis of the interactions in the three-component system Bax, Bcl-xL and cBid, both free in solution and in artificial membranes. The aim was to understand the role of these proteins in the "hole-burning" of the mitochondrial membrane during apoptosis.

The paper seems to have been written in a hurry. The findings (which are mainly the associations of the different proteins under various conditions, as measured by FCS and FCCS) are listed in a lengthy results section, without attempting to summarize them into some scheme that helps the uninitiated reader understanding what is the main statement here.

We have now rewritten the manuscript to improve it and make it clearer. We have added summarizing sentences to the different results sections and small schemes into some of the Figures (Figure 1D and 4B and 8A), which should provide visual guidance about the monomeric or multimeric protein species that can be found in solution or membranes. All our results, together with literature data, are summarized in the model shown in Figure 8. Moreover, a native speaker had a look at the manuscript.

The reader should not be left to the literature for understanding what scanning FCS actually is. The method, in particular two-focus scanning FCCS - and the procedure to correct for membrane movement during the measurement - must be briefly explained. It is certainly not textbook knowledge.

Following the reviewer's suggestion, we have now added basic information in the main text about the FCS variants used in this study. We have also complemented our Supp. Information (and Supp. Figures) with a more extensive introduction to scanning FCS, including also the two-focus, two-color version. Moreover, we include now, also in the Supplementary Information, all the necessary controls to perform quantitative FCCS experiments.

I would take exception to the claim that the system studied here is a "minimal interaction network" using a "synthetic biology approach". A network would consist of meshes, three partners would at maximum form a triangle, not really a network. There is also no "synthetic biology" in here, just classical biochemical / biophysical analysis of isolated protein-protein interactions. The authors should be careful with such buzzwords, which often are used only to make a study sound more "hip". Also the word "recruiting" should be replaced by "binding" unless there is any active process going on that is different from binding (by random diffusion).

Here we studied the interaction of five basic components (p7 and p15 of cBid, Bax, Bcl-xL and the membrane) that form several competing, reversible complexes, giving rise to more than 10 different species. We think this is more complex than a triangle and, likely, also the underlying reason why the regulation of such a simplified system is not well understood yet. For us "network" seemed to be the best description for this minimal interactome (see Figure 8).

We agree with the reviewer that the term "synthetic biology" has been used very differently in literature. Taking the reviewer's comment into consideration, but we have changed it now to a "bottom-up approach".

The term recruiting was used here on purpose. The membrane insertion of Bax and Bcl-xL needs a catalyzer und is therefore not a simple binding process. The term recruiting has been used before in literature to describe this process and seemed therefore appropriate in the context of our manuscript.

In general, the language is typical of current "scientific paper jargon". As an example "multitude of" (p.4 last par.) could be substituted by "many" without loss of information, making lighter reading. "Bottom up synthetic biology approach" (same par.) conveys no information whatsoever, could be deleted. "Intriguingly, strikingly, dramatically, interestingly" and other overexaggerations should be deleted wherever they occur.

We acknowledge that the reviewer has a preference for a different scientific writing style and respect that. We have tried to consider the suggestions made for improvement where possible, but in the end it is our manuscript and a certain level of freedom in the way the findings are communicated should be acceptable. We have tried too to minimize what the reviewer calls "overexaggerations" in the text, but we believe that this kind of adverbs are in some cases useful to draw attention to certain parts of the work that we consider of special interest.

In several places there are incomplete phrases, sometimes rendering the text incomprehensible; e.g. p.7 last par.: "Next we probed the competitions between the Bcl-xL can form homo-dimers and hetero-complexes with cBid". ???

We apologize for this. We have carefully updated the manuscript and involved a native speaker in the corrections to avoid language mistakes.

In spite of these language shortcomings, the Results section is well described, although the amount of detail presented without attempting to consolidate the findings into a framework makes it very difficult to read.

Following the reviewer's suggestion, we have now added summarizing sentences at the end of the different sections and graphical schemes of the species detected (new Figure 1D and 4B).

At any rate, the paper should be revised by a native speaker and good scientific writer.

A native speaker revised the new version of the manuscript.

Technical issues

We thank the reviewer for raising some important questions about technical issues related to FCS and the necessary controls for our quantifications. We have added an additional description of the

methods into the Supp. Information that addresses all these points. This part contains a brief review about the FCCS methods used in our study and explains critical issues when doing quantitative FCCS measurements and how we dealt with them. Additionally, we have addressed all questions made by the reviewer here point-by-point.

1. The spectral ranges for the detection channels are nowhere indicated. This is critical for checking the consistency of the crosstalk/background figures given.

This information is now given in the manuscript. The detector in the "green" channel is detecting light from 505 to 540 nm, the detector in the "red" channel detect light with wavelength >655nm.

2. p.5, par.2: "the baseline for CC was below 2% (data not shown)" - this data MUST BE SHOWN!

We now added Supp. Figure 1C to illustrate the CC /Cross talk between the free dyes, which is also explained in the Supp. Information.

Supplementary information has a lot of space. Why is the threshold for (qualitative) interaction set to 3%, is this based on some statistical criterion or just arbitrary?

We apologize that this part was not well explained in the initial manuscript. We now added information about the statistical criteria we used.

In solution and membranes, we set the initial threshold to detect interactions roughly to a mean $\mu \pm 2\sigma$ or a 95% confidence interval. This was calculated by measuring FCCS between free Alexa488 and Alexa655 shown in Supp. Figures 1C. With Atto488 and Alexa655 we had similar results (even slightly less cross talk).

From this data we calculated the mean (μ : 1.4) and standard deviation (σ : 0.7) of the "background" cross-correlation of solution measurements with the dyes used. This data come from a concentration range between 5-200 nM and the $\mu \pm 2\sigma$ is 2.8 % CC, thus below 3%. Below and above the indicated concentration range the background CC rose, so that experiments were best performed in this range. This information is now given in the main text. As shown in Supp. Figure 7B the cross talk and noise level in membranes is much higher. Here we only consider mean %CC values over 20% as interaction ($\mu \pm 2\sigma = 19\%$).

We would like to point out that the two pairs, Bcl-xL-Bcl-xL and cBid-Bcl-xL, for which we detected cross-correlation in solution showed values much higher than 3% (see Figure 2D and 3B).

3. No positive control for 100% cross correlation is given. This could be easily checked with an A488 / A633 double labeled probe.

For cross correlation measurements one needs to be aware that the detection volume of red and the green detection channel do not perfectly overlap, as the size of the detection volume is directly proportional to the wavelength of the laser light and they may not be perfectly aligned. Following the reviewer's suggestions, we performed experiments to determine the maximal cross-correlation that can be measured in both channels of our setup. These measurements are now shown in Supp. Figure 1A.

As suggested by the reviewer an ideal sample to do such measurement would be a stable molecule to which our red dye (Atto 655) and our green dye (Atto488 or Alexa 488) are attached with 100% labeling efficiency and that contains no free dye. Moreover, such molecule would need to be soluble in buffer and diffuse. Unfortunately, it was impossible for us produce such a sample or to buy it. Maybe the reviewer is aware of the common use of labeled DNA as test samples. We have used those

routinely in our lab to check for cross-correlation performance. However, those samples are not ideal either and showed less cross-correlation than the sample introduced here.

We used egg-PC liposomes (LUVs) labeled with the two lipidic dyes DID (excited by the 633 laser, emission maximum >655 nm) and DIO (excited by the 488 laser, emission maximum between 505-540 nm), which should contain several probes per particle. The vesicles were passed through a 200 nm filter to be more homogenous in size. We detected >90% CC in the red channel and >75% CC in the green channel (Supp. Figure 1A). These results provide an estimation of the upper limit in the cross correlation values due to imperfect overlap of the detection volumes. Nevertheless, as we are aware that this sample is not perfect either, we decided not to calibrate our cross-correlation data to these maximum values. As a result, we certainly underestimate the extent of complex formation, but this does not affect the conclusions of our results (all samples are affected in the same way). But we agree with the reviewer that this is an important aspect in the quantification, which introduces a source of systematic error in the calculation of K_D (see below) and therefore we have now explained this point with detail in the text.

4. Since the positive control is missing, all statements regarding binding are necessarily qualitative. I do not understand how a detailed qualitative analysis of a complex thermodynamic model is possible in this case.

As explained above the extent of complex formation is slightly underestimated in our calculations, however it does not affect our conclusions. We understand that the reviewer means here that this underestimation precludes a detailed quantitative analysis.

We agree with the reviewer that a detailed quantitative analysis including a thermodynamic model of the minimal Bcl-2 interaction is not possible in our case. The most important problem preventing us from providing a real quantitative, complex thermodynamic model is the competition between the different reactions found in solution and in membranes as well as the transition between both environments, which we can detect, but not completely quantify including kinetics. To do so we would need three or even four “color” experiments, performed simultaneously in solution and membranes and correlating between all conditions. Although we are working on the improvement of data quality, unfortunately, this is at the moment technically impossible.

5. This is strikingly evident in Figure S1 (homodimerization). How can this graph be produced when the 100% binding limit is not known?

We have discussed this issue also above. The K_D value in Figure S1 (now S2) is slightly overestimated. However, as explained in the text the real K_D is anyhow much lower like ~10nM or below, as inferred from the mathematical modeling. Thus the error based on the imperfect spectral overlap does not affect the conclusions of this analysis either. We have nevertheless explicitly explained the lack of correction for partial overlap in our calculations to clarify the situation for the reader.

6. Association data between fluorescently labeled proteins, in particular the long-wavelength fluorophores, are notoriously prone to artifacts due to self-association of the dyes. This point is never discussed nor controlled.

As shown in Supplemental Figure 1B (as example) we did test experiments on the free forms of all dyes used. Actually, we measure FCS and FCCS on them on everyday basis as they are used as reference samples in our lab. The intensity traces, diffusion coefficient and the cross-correlation values never gave any indications for dye self-association under our experimental conditions.

7. How was the purity of the protein labeling checked (labeling degree and NO non-specifically associated fluorophore)?

The reviewer raises a very important point. We have now added an additional part to our Supp. Information to show tests and controls for protein labeling and free dye. For all quantitative FCS measurements, it is necessary to calculate and consider the degree of labeling of the proteins used and the fraction of free dye in the sample.

To estimate the degree of labeling we used absorption spectra as well as mass spectroscopy. For mass spectrometry the proteins were passed through a reverse phase column and we assumed that during the process (and the influence of the solvents used) non-covalently bound dye is released and eluted separately from the column. Thus mass spectroscopy should give us valid information about the degree of labeling. For Bax and Bcl-xL we calculated degrees of labeling between 80 and 100%. cBid was labeled between 60 to 80% (info given in the figure legends).

Non-specifically protein associated fluorophores can be indeed a problem in FCS, if they are released under conditions that can be not controlled. In principle, it is not important, whether a dye is covalently or non-covalently attached to a protein as long as the dye is not released during the experiment or the experiment preparation and as long it is not affecting protein function. However, in water-based buffers we also had dye non-covalently attached to the protein and “free” dye in equilibrium. Thus, the dilution before the actual FCS measurement will release some dye from the protein. This issue is only important for measurements in solution (see below).

We optimized the protein purification to minimize the amount of free dye, but we could not remove it with 100% efficiency. To calculate the actual fraction of free dye after dilution, we measured FCS and analyzed the data using a two-component fit, in which the diffusion time of the free dye was fixed. The diffusion time of the free dye is known from the reference measurements we do on the free dyes prior to the actual measurements on proteins. The two-component fit allowed us to estimate the fraction of labeled protein and free dye (Supp. Figure 1D; showing also a one-component fit for comparison). We included this information into our data quantification. For Bax and Bcl-xL the amount of free dye was ~20%, for cBid the amount was a bit higher (~ 20-30% for the green version and 30-40% for the red). We would like to point out that at nanomolar concentrations a fraction of the proteins stick to the chamber walls and glass (despite our attempts to block those surfaces). cBid_R stucked most of all proteins used here, which could be the reason for the higher fraction of free dye in these samples. As cBid_R was the most difficult sample, we only used it here for homo-dimerization studies.

Luckily, there is no detectable binding of the free dye molecules to the membranes used in our experiments, as we confirmed before (Bleicken et al. Biophys J. 2013;104(2):421-31.). As a result, we do not need to consider free dyes in any of the membrane measurements.

8. p.8 last par.: the whole explanation of the increase of CC amplitude upon addition of cBid is completely obscure to me. Why would the formation of a cBid/Bcl-xL heterodimer lead to an increase of the CC amplitude between Bcl- xLR and Bcl-xLG?

That result was also very surprising for us. It can be explained by a very stable Bcl-xL homo-dimer with a low exchange rate, so that during the time of our experiment monomer exchange within dimers hardly takes place. In that case, it is possible to have a very high concentration of Bcl-xL homo-dimers, but only very few two-colored complexes, as without monomer exchange the sample will contain mainly pure green and pure red homo-dimers. In line with this idea, the diffusion coefficients we detected for Bcl-xL in solution (Figure 2A) were in line with a high concentration of homo-dimers (see Figure 2). Based on our data, we interpreted that the addition of cBid acted like a catalyzer of the

exchange process. Reversible hetero-complexes of cBid and Bcl-xL were formed and co-existed with Bcl-xL homo-dimers. The presence of cBid thus raised the exchange rate of Bcl-xL monomers within the dimers. Alternatively, as suggested in scenario 3, cBid could induce the formation of hetero-trimers that released Bcl-xL monomers at one point. We adapted the text explain this in a more clear way.

Conclusion

In general, the paper reports on some interesting new findings, which lack, however, necessary controls and explanations. However, while the problem studied here is certainly interesting for people working in the field of mechanism of apoptosis, it seems to be too specialized to be of interest to a large multidisciplinary audience. I would not recommend publication in NCOMMS.

We hope that the controls and explanations provided above have convinced the reviewer that our new findings are based on carefully performed experiments and a solid data analysis, where we avoided conclusions based on calculations beyond our experimental limitations.

In addition, we would like to point out that the regulation of apoptosis by the Bcl-2 network remains one of the main questions in the field. The use of the Bcl-2 proteins as targets in cancer treatment, with the first small molecule inhibitor targeting protein/protein interactions, precisely between Bcl-2 members (Navitoclax), approved by the FDA early this year also shows the relevance of our work in cancer research and biomedicine. In agreement with that, all previous models of Bcl-2 interactions were published in highly recognized journals with a broad audience. For example:

Llambi F, *et al.* A unified model of mammalian Bcl-2 protein family interactions at the mitochondria. *Mol Cell* **44**, 1-15 (2011). 177 times cites according to “webofknowledge.com”

Willis SN, *et al.* Apoptosis initiated when BH3 ligands engage multiple Bcl-2 homologs, not Bax or Bak. *Science* **315**, 856-859 (2007) 706 times cites according to “webofknowledge.com”

Kuwana T, *et al.* BH3 domains of BH3-only proteins differentially regulate Bax-mediated mitochondrial membrane permeabilization both directly and indirectly. *Mol Cell* **17**, 525-535 (2005). 720 times cites according to “webofknowledge.com”

Letai A, Bassik MC, Walensky LD, Sorcinelli MD, Weiler S, Korsmeyer SJ. Distinct BH3 domains either sensitize or activate mitochondrial apoptosis, serving as prototype cancer therapeutics. *Cancer Cell* **2**, 183-192 (2002). 972 times cites according to “webofknowledge.com”

Leber B, Lin J, Andrews DW. Still embedded together binding to membranes regulates Bcl-2 protein interactions. *Oncogene* **29**, 5221-5230 (2010). 208 times cites according to “webofknowledge.com”

Chen HC, *et al.* An interconnected hierarchical model of cell death regulation by the BCL-2 family. *Nat Cell Biol*, (2015). 17 times cites according “webofknowledge.com”

Beyond this, we are convinced that the quantitative analysis of an interaction network including protein/protein interactions within membranes is also of great interest for quantitative biology, specially the fields of cell signaling and biophysics.

Reviewer #3 (Remarks to the Author):

Bleicken et al. present fluorescence cross correlation spectroscopy (FCCS)-based studies in solution and in a lipid membrane with mathematical modeling to systematically quantify the interactions within a minimal Bcl-2 network using representative proteins of the Bcl-2 family such as cBid, Bax and Bcl-xL. The study would require an independent review of the mathematical modeling and FCCS methodology, but also data on the purity, size and activity of the fluorescence labeled proteins, which are not provided.

*To address the reviewer's concern about protein purity, we now added an exemplary SDS PAGE image from cBid, Bax and Bcl-xL after labeling with the fluorophores to the Supplemental Figures (Supp. Fig. 1E). The protein activity of unlabeled and labeled proteins used here was addressed previously in (Bleicken et al. Biophys J. **104**(2):421-31. (2013) and Subburay et al. Nat Commun. 6:8042 (2015)). We now added additional explanations to the main text and the Supp. Info to clarify these points.*

Nevertheless, the study largely confirms interactions and conclusions previously presented in cell biology, structural biology and biochemical studies by other groups (several of these studies are cited in the paper).

We disagree with several points raised by the reviewer. We would first like to point that we were interested in obtaining a (more) complete picture of the interactions between full-length Bcl-2 proteins in solution as well as in membranes, including their recruitment to membranes and their retro-translocation back into solution. Therefore, we needed to monitor (and when possible compare) all the processes in one system. So it is actually essential that some of our results are necessarily confirming already published work and we refer to that in our text. But to our best knowledge, such a systematic, quantitative and integrative approach was never done before. Considering how controversial some of the published data are, we are convinced that our findings provide important new insight that significantly progress our knowledge about the Bcl-2 signaling network. Below, we address each point raised by the reviewer in detail.

*We would also like to mention that in order to get an integrated, quantitative view of the Bcl-2 network, the direct comparison of quantitative data gained by different techniques in different sample system is not possible. This illustrated below for the tBid-Bcl-xL pair in solution for which K_D 's between 12 nM (Certo M, et al. Cancer Cell **9**, 351-365 (2006) to 350 nM (Kuwana T, et al. Mol Cell **17**, 525-535 (2005)) were published. In this context, it is indeed essential to re-assess all these interactions in a unified system with the same technique.*

For example:

- the associations of the proteins in solution and in membranes are known

*We agree with the reviewer that the interactions between Bcl-2 proteins in solution has been extensively studied, also quantitatively. However, one key aspect in this regard is that most of these studies were performed with truncated proteins and with BH3 peptides or chimeric proteins. For example, one of the best studied interaction pairs within the Bcl-2 protein family is cBid and Bcl-xL in solution. It was studied in many papers (e.g. by us in Garcia-Saez et al. Nat Struct Mol Biol **16**, 1178-1185 (2009); Llambi F, et al. Mol Cell **44**, 1-15 (2011); Certo M, et al. Cancer Cell **9**, 351-365 (2006); Kuwana T, et al. Mol Cell **17**, 525-535 (2005); Hockings C, et al. Cell Death Dis **6**, e1735 (2015); Chen L, et al. Mol Cell **17**, 393-403 (2005); Kim H, et al. Nat Cell Biol **8**, 1348-1358 (2006); Aranovich A, et al. Mol Cell **45**, 754-763 (2012); Willis SN, et al. Science **315**, 856-859 (2007)). It is important to note that the quantitative results strongly vary between the studies, with K_D 's between 12 nM (Certo M, et al. Cancer Cell **9**, 351-365 (2006) to 350 nM (Kuwana T, et al. Mol Cell **17**, 525-535 (2005)). As mentioned*

above, most of these studies were not performed with the full length proteins, which clearly affects the affinities (e.g. shown by (Hockings C, et al. *Cell Death Dis* **6**, e1735 (2015)). Therefore, even for this intensely studied pair, the new data presented here provides new information.

Moreover, one highly neglected observation in the apoptotic field is that full length Bcl-xL forms homo-dimers in cells (Jeong SY, et al. *Embo j* **23**, 2146-2155 (2004)), while the C-terminal truncated version of Bcl-xL does not form a detectable amount of homo-dimers (Jeong SY, et al. *Embo j* **23**, 2146-2155 (2004)). This is a key issue because it affects all previous studies of hetero-dimers including a Bcl-xL molecule: homo-dimerization competes with the hetero-dimerization process and should be considered decoded in order to understand the Bcl-2 interaction network. To our knowledge this issue is taken into account here for the first time for all three full-length proteins in solution and in membranes. As we discuss in the manuscript, we provide a potential explanation how the stable Bcl-xL dimer detected here may be relevant in terms of enabling the Bcl-xL induced retro-translocation of Bax into the cytosol (Mode 0 interaction) without an additional energy source, which remains an important question in the field.

These examples pinpoint how our individual findings for the interactions between Bcl-xL, Bax and cBid in solution shed new light on the orchestration of the Bcl-2 network. If we consider now the interactions in membranes, the situation is much more critical. It is certainly known that Bcl-2 proteins interact in membranes, but the quantitative information in the literature is very limited. As discussed below, we have indeed contributed in our previous work to understand Bcl-2 interactions in membranes. We also agree with the reviewer's later comment, that the groups of David Andrews and collaborators have made key contributions to understand membrane-embedded Bcl-2 proteins. However, the matter is by far unsettled and many controversies remain, as for example the relative importance of these interactions in the membrane (see below). Here, we have also identified homo-oligomers of Bcl-xL in the membrane for the first time.

Another important aspect is that those studies were done before the labs of Richard Youle, Frank Edlich and Andrew Gilmore established that Bcl-2 proteins do not only translocate to MOM membranes, but also shuttle back into the cytosol (e.g. Schellenberg B, et al. *Molecular Cell* **49**, 959-971 (2013); Edlich F, et al. *Cell* **145**, 104-116 (2011)). Thus, there is a steady state between soluble and membrane bound which should be considered to understand Bcl-2 interactions in membranes. This finding is critical in the sense that it becomes necessary to experimentally separate the interactions in the membrane and to exclude interactions in solution from the analysis. This was not done in Andrews work, because it is impossible to do that by conventional FRET as well as most other techniques, but it is possible with the scanning FCCS technique used here. In this sense, our data are unique and allow to separate the interactions between Bcl-2 proteins in solution and in membranes, and to quantitatively compare association preferences, which as we show in our manuscript, is critical to understand the Bcl-2 network.

- affinities are also established to be higher in the membrane than in solution

We agree with the reviewer that it is known that the affinity between cBid and BclxL is higher in the membrane than in solution (as we showed some years ago (Garcia-Saez et al. *Nat Struct Mol Biol* **16**, 1178-1185 (2009)) and that Bax molecules require the membrane to self-assemble (Lovell et al. *Cell* **135**, 1074-1084 (2008)). However, the key open question that our study solves is the preference of the interactions between Bax, Bcl-xL and cBid within the membrane. Another key finding is that the C-terminal transmembrane helices play a key role in determine the hierarchy of membrane interactions (see below).

- spontaneous Bax activation by temperature, autoactivation mechanism and oligomerization in the membrane are also established

We agree with the reviewer and we do not claim to that any of those three processes are novel findings of our work. We also agree that Bax oligomerization in the membrane is relatively well established. However, even this point was recently challenged (Xu et al. Cell Death Dis 4, e683 (2013), Kushnareva et al. 10(9), e1001394 (2012)), which reflects how many “established observations” in the apoptosis field are still controversially discussed, because we still do not completely understand the mechanisms inducing and preventing MOMP.

However, we believe that the molecular mechanisms involved in Bax auto-activation are not yet understood and that there is still strong controversy in the field regarding the ability of Bax to activate itself. One has to be careful with the term Bax auto-activation as it is used differently in literature. Spontaneous Bax activation by temperature is indeed known and we do not claim that this is a new finding. We exploit it in our chemically controlled system to perform experiments in the absence of cBid.

Some groups (e.g. Tan et al (J. Biol. Chem.281(21):14764-75 (2006)) used peptides of Bax helix 2 or helix 2 and 3 to show that they can act like activator BH3-only protein peptides and indeed that can be called an established mechanism. In addition, they reported that membrane-bound Bax activated the pore activity of soluble Bax molecules followed by experiments of content release from liposomes. We acknowledge this study in our manuscript, which was indeed the basis for the experiment demonstrating that membrane bound Bax recruits soluble Bax molecules to the GUVs shown in figure 7. Although we point out again that the most intriguing finding that we report in this respect is that membrane-bound Bax can recruit Bcl-xL to the GUV membranes.

Besides, in recent work, O’Neill et al. (Genes Dev 30, 973-988 (2016)) produced cells in which almost all known Bcl-2 proteins were knocked out (except BOK) and they reintroduced Bax back into these cells. Surprisingly in those cells it was still possible to activate Bax by pro-apoptotic stimuli and the authors therefore conclude that Bax is auto-active, or constitutively active. This is an indirect approach as in principle Bax could be also be activated by a so far unknown factor. One could think some Bax monomers could activate others by helices 2 and 3 but why should they be accessible? They are involved in Bax compact globular fold and should not be available without an extra trigger (Suzuki et al. Cell (2000)). Without an explanation to this we can not claim to understand Bax-autoactivation.

Another recent publication (Zhang et al. Embo j 35, 208-236 (2016)) describes “auto-active” Bax mutants, which in this case means that the proteins permeabilize mitochondria without an activator BH3 only protein present. Interestingly the “auto-active” Bax mutants L76C and V110C are in the dimerization domain interface of Bax, thus they will likely affect Bax conformational change, and may therefore act similar to activator BH3-only protein peptides. This idea is strengthened by Zhang et al. additional finding that in healthy cells Venus-fused-Bax-mutants containing L76C are localized to mitochondria without directly inducing cell death (like the G179I mutant did). Thus Bax L76C likely has a non-active mitochondria bound conformation that may act like an activator BH3-only peptide.

With these arguments we hope to convince the reviewer that, based on current knowledge, one can only speculate whether the Bax “auto-activation” mechanisms described by the articles mentioned above are one single mechanism or different mechanisms. Therefore the Bax “auto-activation” mechanism (or mechanisms) are far from being understood.

We also think that the fact that membrane-bound Bax not only can recruit soluble Bax molecules to the membrane, but also soluble Bcl-xL molecules supports a common mechanism for both processes which was unanticipated. We have now emphasized this in the revised manuscript.

- recruitment of Bax and Bcl-xL in the membrane by cBid is previously shown

We agree with the reviewer and we did not intent to state that this in new. Indeed we wrote: "It is well established that cBid promotes the association of both Bax and Bcl-xL to membrane containing negatively charged lipids like cardiolipin^{17, 18, 19, 49}."

However, here we show that membrane-inserted Bax can recruit soluble Bax to the membranes as discussed above, and, most importantly, that membrane-inserted Bax is even able to recruit soluble Bcl-xL to the membrane. This is an unexpected finding that sheds new light into the molecular mechanism of Bax inhibition by Bcl-xL. Moreover, it demonstrates that the process by which Bcl-2 proteins are recruited to membranes is more complex than previously thought and deserves further investigation.

- inhibition of MOMP by Bcl-xL using a higher affinity to cBID than Bax is established

*We disagree. The probably most cited publication in regard to this is Llambi et al. (Mol Cell **44**, 1-15 (2011)) and it claims just the opposite. The reason for this conflict is that Llambi et al. used truncated Bcl-xL, which as we show here, strongly affects the Bcl-2 protein interactions.*

- dissociation of activated BAX in the membrane by Bcl-xL is established

*To our best knowledge this was a working hypothesis only recently demonstrated by us (Subburaj et al. Nat Commun **6**, (2015)) using a single molecule imaging. We do not claim that our results are new in this regard, but we are convinced that it is worthy showing the FCS data supporting these results, because they have been obtained with second independ method.*

- the key role of the c-terminal helix of Bcl-xL in regulating interactions is known

We strongly believe that the role of Bcl-xL's CT is underestimated, as most studies still work with the C-terminal truncated form with the excuse that it is "only" the membrane anchor and neglecting its regulatory role in protein interactions. This leads to important controversies in the field, like in the case of the work by Llambi et al mentioned above, indicating that the matter is clearly unsettled. In our opinion, it is therefore critical to emphasize the importance of using full-length proteins and to show how the CT truncation changes the hierarchy of interactions.

Also similar studies performed by the group of David Andrews which has found similar results in reconstituted membranes using FRET and biochemical readouts (Billel LP Plos Biol 2008, Lovell JF Cell 2008, Tan C JBC 2004, Shamas-Din A JBC 2013) Therefore, to mention that the "relative importance of the BCL-2 family interactions are poorly understood" is an overstatement.

We have now rewritten the manuscript and changed that sentence to "due to conflicting results, the relative importance of the Bcl-2 family interactions remains unsettled".

I understand that FCCS provides a fresh insight into the individual interactions of proteins compared to more established approaches but I also strangle to understand the impact of this study in more physiological context, where proteins can undergo modifications, receive stress signals and their concentrations change, which in several cases these phenomena cannot be fully reproduced by the recombinant proteins. Therefore, this study is more appropriate for a focused biophysical journal.

We agree with the reviewer that in cells the situation is much more complex and the proteins can undergo modifications, or the protein environments change after certain stress, etc. As a result, an intrinsic disadvantage of reconstituted systems is that they only partially mimic the situation in vivo.

But precisely this is their strength too, as these approaches reduce the complexity of the system, so that it becomes possible to study biological processes, in this case the Bcl-2 signaling network, with a depth that is not feasible in the cellular context. This happens at two levels: on the one hand, the advanced biophysical techniques that we used in this study to quantify interactions between Bcl-2 proteins in membranes cannot be applied to the mitochondria of cells. On the other hand, in the cellular environment there are a number of both unknown and known but uncontrolled factors that affect the behavior of the system and limit the knowledge that can be extracted from the experimental data. Reconstituted systems offer in contrast chemically controlled conditions that allow changing parameters like lipid composition, protein concentration, pH, temperature, etc. almost at will. As a result, reconstituted systems have proved extremely useful to learn about the functioning of many biological systems over the last decades. One critical aspect here is that the simplified, chemically controlled system faithfully reproduces the concrete features of the aspect of the biological system under investigation. In this sense, the minimal Bcl-2 network studied here is based on extensive functional characterization of the reconstituted components individually and as a system that we did in our previous work and that also builds on the literature. A second important aspect is that the conclusions from reconstituted systems should only be extrapolated with extreme care to the physiological context. We have now emphasized this aspect in the revised version of the manuscript.

*Finally, we would like to provide concrete examples of how simplified systems like the one used here have proven extremely useful to our understanding of Bcl-2 proteins. Many of hallmark articles about Bcl-2 protein interactions, Bcl-2 protein induced pore formation, Bcl-2 protein structure and the molecular mechanisms how these proteins function reported work not done in cells, but in model membrane systems comparable to ours (e.g. Czabotar et al. *Cell* **152**, 519-531 (2013); Gavathiotis et al. *Mol Cell* **40**, 481-492 (2010); Bleicken et al. *Molecular Cell* **56**, 496-505 (2014); Billen et al. *PLoS Biol* **6**, e147 (2008); Llambi et al. *Mol Cell* **44**, 1-15 (2011); Kuwana et al. *Cell* **111**, 331-342 (2002); Suzuki M, *Cell* **103**, 645-654 (2000); Lovell et al. *Cell* **135**, 1074-1084 (2008); Gavathiotis, et al. *Nature* **455**, 1076-1081 (2008); Sattler et al. *Science* **275**, 983-986 (1997); Montessuit et al. *Cell* **142**, 889-901 (2010)).*

Reviewer #4 (Remarks to the Author):

In this well performed and clear study, Garcia-Saez and colleagues further develop our current knowledge of the BCL-2 family protein regulation of apoptosis, taking into account interactions within membranes. Through very careful experimental design and execution in reconstituted systems (LUVs), the authors suggest that Bax can auto-activate (at physiological temperature) providing a positive feed-back mechanism that requires regulation by Bcl-xL. Bcl-xL regulation comprises three distinct steps identified by the authors: it translocates Bax back into solution, binds BH3 only proteins (cBid) limiting their interaction with Bax and binds Bax oligomers limiting their size. Each of these aspects is very important for our understanding of the process of MOMP, and a 'unified' model has been developed by the authors.

We thank the reviewer for the positive evaluation of our work and for acknowledging the relevance of the findings reported here.

Minor criticisms:

1. In the Discussion, the authors should discuss more carefully the limitations of their current system in terms of their reconstitution assays used.

Following the reviewer's suggestion, we have now detailed explanations of the limitations of our reconstituted system, which can be found when we introduce the system in the results section.

2. The authors should also discuss how this model relates to Bax (and BCL-xL) activities OTHER than apoptosis regulation such as Ca²⁺ fluxes at the ER and fusion/fission regulation (which could be their primary functions evolutionary).

We thank the reviewer for this comment. We have now added additional information on the non-apoptotic functions of Bcl-2 protein and the potential implications of our results to these functions.

Reviewers' Comments:

Reviewer #1:

Remarks to the Author:

This study provides important independent verification of the popular biological finding of retrotranslocation of BAX from membranes by BCL-xL using biophysical approaches, which further indicates that retrotranslocation can occur without additional cellular proteins. In contrast to cell-based studies, this study also uses full-length proteins that are not tagged with GFP or other large fluorophore, and takes into account the effects of homodimerization on the interactions between BCL-2 family members. This study brings together information regarding soluble AND membrane interactions to test the different hypothesized "Modes" of action leading to membrane permeabilization/MOMP; the results support different features of each of these modes [cBid sequestering & neutralization by BCLxL in membranes & in solution supports mode 1; BAX-BCLxL complexes in membranes only with heat supports mode 2; in membranes, the cBid and BCLxL complex does not include p7, whereas it does in the soluble forms; homodimerization of BCLxL could be the driving mechanism behind retrotranslocation (i.e. mode 0 inhibition)]. However, the text needs modification to clearly articulate these new findings. There is also duplication of previously published data that needs to be removed and/or more clearly articulated.

Specific comments:

1. Although the authors cite their 2013 Biophys J paper when referring to Figure 4A on page 11, it is inappropriate to republish the identical graphs (the left two panels for BAX and Bcl-XL homodimers in Figure 4A are the exact same as published Figure 4A and 4B in Bleicken et al., Biophys J., 2013). In addition, the legend for Figure 4E indicates that some of these data were also published previously, but exactly how much new and published data were merged here was not explained. Any previously published data needs to be removed, or in rare cases when combined with substantial new data, the overlap must be made crystal clear.
2. The writing style sells short the accomplishments reported here. The text does not do a good job clarifying the unique contributions of this study, and does not adequately explain exactly where these findings agree or disagree with previous reports that use different strategies. The abstract and final paragraph of the Introduction should be re-written in this regard to avoid irritating readers with language that simply restates what has already been reported using different strategies.
3. Prior work on Bcl-2 family transmembrane domains needs to be better related to the current study, e.g. the works of Andreu-Fernández et al. on the interaction between C-terminal TMDs in membranes are not mentioned here. A second example is the study of Sung et al. (Structure, October 2015) using DEER and other biophysical techniques to identify a soluble oligomeric structure; please provide specific comments to help resolve or address this discrepancy with the current manuscript.
4. This sentence in the Introduction: "So far, the interactions between Bcl-2 proteins have been analyzed one at a time" (page 3), should be deleted as this is not the case. In the same paragraph, delete or rephrase: "we provide the Bcl-2 interactome", as only 3 proteins are studied here. Simply put, the language in many places to not sufficiently articulate.
5. The first sentence of "Results" is false and should be deleted; this point can be reworded in a more specific and non-ambiguous manner. In the next sentence, the lack of posttranslational modifications could be viewed as a problematic limitation to the biophysical methods applied in this study (perhaps the authors meant that fluctuations in protein modifications in cells could introduce confounders). New paragraph 2 of Results is at least partially redundant with the first paragraph and should be removed or reworked with the opening paragraph. At the start of the 3rd paragraph of Results (which arguably should be the 1st paragraph of Results), state the specific objective for using FCCS – state what the very specific point does this experiment seek to demonstrate that is not already known.
6. The Introduction states that there are 5 models for MOMP, but only 3 are described or distinguished (clarify whether there are 3 main models labeled with 5 names, or elaborate if the authors wish to distinguish all 5 models including those lumped together here). Similarly, the authors had proposed a 'fourth' [misspelled, page 3, line 4] Mode of action (Mode 3?), but do not

adequately explain their new Mode or its relationship to Modes 0-2. In several places the authors indicate that they "disentangle" the events that occur, but do not explain what details are being disentangled.

7. In each section of the Results section, clarify what exactly is novel, confirming, or dissenting about the new results compared to published findings.

8. Contrary to the authors' rebuttal, the number of technical replicates and independent experiments performed still need to be provided (preferably in the legends) for figures 1, 2, 3, 7, S1, S3, S4. The authors also neglected to indicate whether the error bars in Figure 2 are SEM or SD. Please include statistics to show significance for the effect of BCLxL on reducing cBID diffusion in figure 2C.

9. The description of the modeling in Figure 3 (and S3 and S4) is difficult to decipher and interpret.

10. In the text, some of the binding characterizations were rather qualitative (i.e. cBid and BCLxL bind with high affinity; cBID and BAX bind with low affinity); assign values when asking the reader to compare figures to make it easier to read. For example, on page 13, when describing the affinity of cBid/BAX vs. cBid/BCLxL heterodimers, indicate the mean values for each when asking the reader to "compare Figure 6A with Figure 5A, mean 40% vs. 60%).

11. Figure 2D – what are the red and black dots (black dots could be shadows appearing on the [BCLxL]/[cBID/BCLxL] axis).

12. Figure 4C – the top two rows are labeled identically (same for the bottom rows) but differ in appearance; explain in the legend [would there be a BCLxL signal present if the 1st and 3rd rows were as highly exposed?]. Clarify the statement that BAX-XL dimers are only observed after heat treatment (as in Figure 4G), because the GUVs in Figure 4C seem to imply that both heat and Bid can bring BCL-XL to the membrane. Does this actually distinguish membrane targeting of BCL-XL with interactions with BAX? Is BCL-xL coming to the membrane via a different mechanism in Figure 4C (recruitment by BID vs direct interaction with BAX in the heat treated GUVs)? It was unclear how the conclusion on page 14 was derived – how one BCLxL molecule is able to inhibit more than one molecule of BAX.

13. Figure 7D needs a key for the various colored dots.

Reviewer #2:

Remarks to the Author:

The revision has greatly improved the readability of the paper and considered most of my technical comments. The argumentation as to why the "real" Kd of homodimers is much smaller than the "measured" one is still quite handwaving, since this system is still not at equilibrium, as the authors concede. Negligible self-association of dyes is not an argument against possible self-association when bound to proteins; experience shows that this is quite often possible. So a word of caution should be given about this in the discussion.

Reviewer #3:

Remarks to the Author:

The authors have made improvements to the updated manuscript providing control experiments and more clarity with the presentation of the data and description of the methodology, results and discussion. These efforts are appreciated. As I previously mentioned and authors also agree, the majority of the interactions with BAX, cBID and BCL-xL here have been previously reported and the model of the simple BCL-2 family network provided in figure 8 is a modification of previous models reported. Previous studies mentioned as hallmark papers provided novel and significant insights e.g. high-resolution structural information and/or biochemical and cellular validation.

The concerning issue here is that some interactions known in the literature are reproduced and some interactions are not reproduced in their set up. However, no effort is done to interrogate or

validate further these observations in a more functional or physiological setup. I will mention one example but there are more that can be said. For example, a BCL-xL dimer in solution and cytoplasm was previously characterized (Jeong et al EMBO 2004), an inactive BAX dimer was crystalized and shown in soluble and cytoplasmic fractions (Garnet et al. Molecular Cell 2016) and dimerization of tBID in solution and membrane as well as oligomerization have been reported with similar approaches (Shivakamur et al. Biophysics J, 2014). The current study was only able to detect BCL-xL dimer in solution but the soluble BAX dimer or soluble and membrane bound tBID dimers or oligomers were not detected. Moreover, recent NMR studies with BCL-xL protein (Yao et al. JMB 2015) that includes the c-terminal helix showed no evidence of the dimer in solution that was previously suggested even at uM concentrations. Because of these inconsistencies with the previous literature it is understood that the current study provides a partial view of the bcl-2 family network and the conclusions here need to be taken with caution.

I am also concerned with the SDS PAGE provided in S1E. The gel shows two bands for labeled-cBID which I presume a cBID monomer and a dimer (but no dimer is detected in solution or membrane here). Moreover, several bands for BCL-xL protein sample and a hardly-detected labeled-BAX despite the high concentrations are shown.

So although this is an interesting study in the field of BCL-2 family and apoptosis, I believe the focus of the study and the impact of the conclusions are narrow and not suitable for publication in Nat. Comm.

Reviewer #4:

Remarks to the Author:

n.a.

REVIEWERS' COMMENTS:

Reviewer #1 (Remarks to the Author):

This study provides important independent verification of the popular biological finding of retrotranslocation of BAX from membranes by BCL-xL using biophysical approaches, which further indicates that retrotranslocation can occur without additional cellular proteins. In contrast to cell-based studies, this study also uses full-length proteins that are not tagged with GFP or other large fluorophore, and takes into account the effects of homodimerization on the interactions between BCL-2 family members. This study brings together information regarding soluble AND membrane interactions to test the different hypothesized “Modes” of action leading to membrane permeabilization/MOMP; the results support different features of each of these modes [cBid sequestering & neutralization by BCLxL in membranes & in solution supports mode 1; BAX-BCLxL complexes in membranes only with heat supports mode 2; in membranes, the cBid and BCLxL complex does not include p7, whereas it does in the soluble forms; homodimerization of BCLxL could be the driving mechanism behind retrotranslocation (i.e. mode 0 inhibition)]. However, the text needs modification to clearly articulate these new findings. There is also duplication of previously published data that needs to be removed and/or more clearly articulated.

We thank the reviewer for the constructive criticism and for highlighting the novelty of our work.

Specific comments:

1. Although the authors cite their 2013 Biophys J paper when referring to Figure 4A on page 11, it is inappropriate to republish the identical graphs (the left two panels for BAX and Bcl-XL homodimers in Figure 4A are the exact same as published Figure 4A and 4B in Bleicken et al., Biophys J., 2013). In addition, the legend for Figure 4E indicates that some of these data were also published previously, but exactly how much new and published data were merged here was not explained. Any previously published data needs to be removed, or in rare cases when combined with substantial new data, the overlap must be made crystal clear.

The reviewer is right and we apologize for this confusion. We have now added curves from different experiments to Figure 4A and adapted the figure and the text to clarify how the data have been merged. In the Biophys J. (2013) manuscript we published data on Bax and Bcl-xL homooligomerization in presence of cBid. In the present study we have analyzed more protein pairs and different conditions, as well as performed additional independent experiments. The analysis of Bax and Bcl-xL homooligomerization has also been included here for clarity of comparison, e.g. in Figure 4E. Because of this, we have referenced the older manuscript and adapted the text to make this clearer.

2. The writing style sells short the accomplishments reported here. The text does not do a good job clarifying the unique contributions of this study, and does not adequately explain exactly where these findings agree or disagree with previous reports that use different strategies. The abstract and final paragraph of the Introduction should be re-written in this regard to avoid irritating readers with language that simply restates what has already been reported using different strategies.

We thank the reviewer for the constructive criticism. We have adapted the text to make this clearer.

3. Prior work on Bcl-2 family transmembrane domains needs to be better related to the current study, e.g. the works of Andreu-Fernández et al. on the interaction between C-terminal TMDs in membranes are not mentioned here. A second example is the study of Sung et al. (Structure, October 2015) using DEER and other biophysical techniques to identify a soluble oligomeric structure; please provide specific comments to help resolve or address this discrepancy with the current manuscript.

We have now added references to those two studies and included them in the discussion.

4. This sentence in the Introduction: “So far, the interactions between Bcl-2 proteins have been analyzed one at a time” (page 3), should be deleted as this is not the case. In the same paragraph, delete or rephrase: “we provide the Bcl-2 interactome”, as only 3 proteins are studied here. Simply put, the language in many places to not sufficiently articulate.

We have rephrased the sentences.

5. The first sentence of “Results” is false and should be deleted; this point can be reworded in a more specific and non-ambiguous manner. In the next sentence, the lack of posttranslational modifications could be viewed as a problematic limitation to the biophysical methods applied in this study (perhaps the authors meant that fluctuations in protein modifications in cells could introduce confounders). New paragraph 2 of Results is at least partially redundant with the first paragraph and should be removed or reworked with the opening paragraph. At the start of the 3rd paragraph of Results (which arguably should be the 1st paragraph of Results), state the specific objective for using FCCS – state what the very specific point does this experiment seek to demonstrate that is not already known.

We have rephrased and shortened the three paragraphs, and some information has been moved to the introduction.

6. The Introduction states that there are 5 models for MOMP, but only 3 are described or distinguished (clarify whether there are 3 main models labeled with 5 names, or elaborate if the authors wish to distinguish all 5 models including those lumped together here). Similarly, the authors had proposed a ‘fourth’ [misspelled, page 3, line 4] Mode of action (Mode 3?), but do not adequately explain their new Mode or its relationship to Modes 0-2. In several places the authors indicate that they “disentangle” the events that occur, but do not explain what details are being disentangled.

The names and references of the 5 models are given on page 3. The newer three, unified, embedded together and the hierarchical models all integrate the older direct and indirect activation models. From our perspective the three models indeed describe similar situations and it is difficult to differentiate between them. This is why we do not appoint differences.

The ‘fourth’ Mode of action is based on our recent observation that Bax and cBid affect membrane curvature in vitro (Bleicken et al *Cell Death Dis* 2016). However, we do not push this finding here due to its limited relevance for this study, and the fact that our finding may not be a forth mode but a consequence of the protein conformational changes described under Mode 1 and 2. We adapted the text and do not mention anymore a fourth mode.

We have adapted the text to clarify the models and rewritten the parts using the word “disentangled”.

7. In each section of the Results section, clarify what exactly is novel, confirming, or dissenting about the new results compared to published findings.

We have now adapted the text to make these aspects clearer.

8. Contrary to the authors' rebuttal, the number of technical replicates and independent experiments performed still need to be provided (preferably in the legends) for figures 1, 2, 3, 7, S1, S3, S4. The authors also neglected to indicate whether the error bars in Figure 2 are SEM or SD. Please include statistics to show significance for the effect of BCLxL on reducing cBID diffusion in figure 2C.

The reviewer is right and we have accordingly added this information to the figure legends, the Supplemental and the material and methods. Concerning Figure 2C, we have added the p-value. We have also added new versions of Figure 2A and C as we realized that the old figures presented a mixture of biological and "technical" repetitions. Thereby these "technical" replicates are data points measured at different protein concentration on the same day. Thus they are not real technical replicates but neither individual experiments. Now we have calculated, for Figure 2A and C, the mean from all repetitions during one individual experiment, which creates smaller error bars. We did not do this originally because the Bcl-xL concentration affected D strongly. However, we have Figure 2B to show this effect. The number of individual experiments is included in the figure 2A and C. However in case of the diffusion coefficient of Bcl-xL-dimers, we have added additional blue bars to indicate in which range the data scatter, taking into account all concentrations and, in case of cBid/Bcl-xL, also all cBid to Bcl-xL ratios used.

9. The description of the modeling in Figure 3 (and S3 and S4) is difficult to decipher and interpret.

Following the reviewer's suggestion, we have now modified the caption of Figures 3, S3 and S4 to improve the clarity of the text.

10. In the text, some of the binding characterizations were rather qualitative (i.e. cBid and BCLxL bind with high affinity; cBID and BAX bind with low affinity); assign values when asking the reader to compare figures to make it easier to read. For example, on page 13, when describing the affinity of cBid/BAX vs. cBid/BCLxL heterodimers, indicate the mean values for each when asking the reader to "compare Figure 6A with Figure 5A, mean 40% vs. 60%).

We have now added this information to the text.

11. Figure 2D – what are the red and black dots (black dots could be shadows appearing on the [BCLxL]/[cBID/BCLxL] axis).

The reviewer is correct. The data in Figure 2D is shown in the red dots. The black ones are the shadows on the axis. For clarification we have now added this information to the figure legend.

12. Figure 4C – the top two rows are labeled identically (same for the bottom rows) but differ in appearance; explain in the legend [would there be a BCLxL signal present if the 1st and 3rd rows were as

highly exposed?]. Clarify the statement that BAX-XL dimers are only observed after heat treatment (as in Figure 4G), because the GUVs in Figure 4C seem to imply that both heat and Bid can bring BCL-XL to the membrane. Does this actually distinguish membrane targeting of BCL-XL with interactions with BAX? Is BCL-xL coming to the membrane via a different mechanism in Figure 4C (recruitment by BID vs direct interaction with BAX in the heat treated GUVs)? It was unclear how the conclusion on page 14 was derived – how one BCLxL molecule is able to inhibit more than one molecule of BAX.

We have adapted figures 4C to clarify this. As explained earlier in the text, we detected that both heat and cBid recruit Bcl-xL (and Bax) to membranes and now we have adapted the text to clarify this point. However we are not able to address whether a different mechanism is involved. We have also adapted the text to explain what suggests that one BCLxL molecule may inhibit more than one Bax molecule.

13. Figure 7D needs a key for the various colored dots.

We have now added this information to the legend.

Reviewer #2 (Remarks to the Author):

The revision has greatly improved the readability of the paper and considered most of my technical comments. The argumentation as to why the "real" Kd of homodimers is much smaller than the "measured" one is still quite handwaving, since this system is still not at equilibrium, as the authors concede. Negligible self-association of dyes is not an argument against possible self-association when bound to proteins; experience shows that this is quite often possible. So a word of caution should be given about this in the discussion.

We thank the reviewer for this constructive criticism. We have now adapted the manuscript to clarify the points mentioned.

Reviewer #3 (Remarks to the Author):

The authors have made improvements to the updated manuscript providing control experiments and more clarity with the presentation of the data and description of the methodology, results and discussion. These efforts are appreciated.

We thank the reviewer for the positive words.

As I previously mentioned and authors also agree, the majority of the interactions with BAX, cBID and BCL-xL here have been previously reported and the model of the simple BCL-2 family network provided in figure 8 is a modification of previous models reported. Previous studies mentioned as hallmark papers provided novel and significant insights e.g. high-resolution structural information and/or biochemical and cellular validation.

The concerning issue here is that some interactions known in the literature are reproduced and some interactions are not reproduced in their set up.

We agree with the reviewer. Currently there are interactions between Bcl-2 proteins for which conflicting results are published. We have indeed discussed these issues in the text. This is also one important reason why we believe the matter is not settled yet and more work is necessary to understand the reasons for the conflicting results. We are convinced that our study here presents an important step forward because it addresses all of the possible combinations of Bax, cBid and Bcl-xL, individually, as pairs and with all three proteins together, in solution and in membranes, with the same experimental approach.

However, no effort is done to interrogate or validate further these observations in a more functional or physiological setup.

We are aware of the limitations associated with reconstituted systems and always have this in mind to avoid over-interpreting our results. Although we are intensively working in extending quantitative approaches to the Bcl-2 proteins also in the cellular context (see Salvador-Gallego et al. EMBO 2016), unfortunately at the moment it is impossible to obtain absolute quantitative data about protein interactions in the organelles of living cells. Indeed, in general, the techniques available to quantify protein interactions, especially within the membrane environment, are very limited and extremely challenging, also in vitro. This is one reason why Bcl-2 protein interactions have been so far mainly studied in solution and why our work provides such a new perspective to the field.

I will mention one example but there are more that can be said. For example, a BCL-xL dimer in solution and cytoplasm was previously characterized (Jeong et al EMBO 2004), an inactive BAX dimer was crystalized and shown in soluble and cytoplasmic fractions (Garnet et al. Molecular Cell 2016) and dimerization of tBID in solution and membrane as well as oligomerization have been reported with similar approaches (Shivakamur et al. Biophysics J, 2014). The current study was only able to detect BCL-xL dimer in solution but the soluble BAX dimer or soluble and membrane bound tBID dimers or oligomers were not detected. Moreover, recent NMR studies with BCL-xL protein (Yao et al. JMB 2015) that includes the c-terminal helix showed no evidence of the dimer in solution that was previously suggested even at uM concentrations. Because of these inconsistencies with the previous literature it is understood that the current study provides a partial view of the bcl-2 family network and the conclusions here need to be taken with caution.

As mentioned above, we agree with the reviewer that conflicting findings about Bcl-2 protein interactions are published. We have mentioned in the text when our data differ from other published results. Especially in the case of Bax and Bcl-xL dimerization in solution, there exist conflicting results.

We can reason about the underlying reasons for these discrepancies. For example we were very surprised that Yao et al. (JMB 2015) detected Bcl-xL as a monomeric protein. However, that might be due to the fact that they were not using the full-length protein, but a truncated version (amino acids 45-84 were deleted) with a His tag. In the case of Bax, Garnet et al. detected Bax dimers in the cytosol, but only in some cell lines and could not explain why. Thus, even based on Garner et al., Bax monomers could be the most physiological form of Bax in solution. The Bid oligomers in Shivakamur et al could be an artefact based on non-fused vesicles or membrane buds on the supported bilayer, which can be an issue in these systems (see Subburaj et al Nat Commun, 2015). Indeed, in Supp. Fig. 7C we also detected cBid homo-

oligomerization, but we confirmed that this was an artefact due to buds on the GUV surface. We have included some of these arguments in the manuscript.

I am also concerned with the SDS PAGE provided in S1E. The gel shows two bands for labeled-cBID which I presume a cBID monomer and a dimer (but no dimer is detected in solution or membrane here). Moreover, several bands for BCL-xL protein sample and a hardly-detected labeled-BAX despite the high concentrations are shown.

Concerning cBid: cBid is the cleaved form of the 21kDa protein Bid. Therefore in SDS PAGE two fragments are visible: The 7kD fragment (p7) and the 15 kD fragment (p15 or tBid; red box in Suppl. Figure 1E version below). On the gel image only two bands are visible that are corresponding to the two fragments. Thus the protein was properly and completely cleaved and there is no indication for dimerization. For clarity the names of the fragments are now shown next to the bands.

Concerning Bcl-xL: The image shows indeed bands in line with monomers and dimers, as well as a faint band around 80 kDa (pink box in Suppl. Figure 1E). As we see stable Bcl-xL dimers in solution, the detection of homo-dimers was not completely unexpected. The faint 80 kDa band could most likely be due to artificial oligomerization in the SDS-PAGE, since for example a similar effect was also detected for BSA (see black box).

Concerning Bax: The protein is only faintly visible on the left gel image due to the low concentration, $\sim 1 \mu\text{M}$ (grey box in Suppl. Figure 1E version below). $1 \mu\text{M}$ Bax corresponds to $\sim 0.02 \text{mg/ml}$ protein, and thus the faint band intensity is in line with the BSA bands (black box), that are shown as a visual guideline for protein concentration. On the right SDS-PAGE image the very same protein from the same purification is shown at a higher concentration (blue box in Suppl. Figure 1E version below).

Copy of supp. Fig 1E for explanations for the reviewer.

So although this is an interesting study in the field of BCL-2 family and apoptosis, I believe the focus of the study and the impact of the conclusions are narrow and not suitable for publication in Nat. Comm.

Reviewer #4 (Remarks to the Author):

The authors have addressed all concerns raised by this reviewers.

We thank the reviewer for the positive evaluation of our work.